# Half-metallic carbon nitride nanosheets with micro grid mode resonance structure for efficient photocatalytic hydrogen evolution

Gang Zhou[1], Yun Shan[1,2], Youyou Hu[3], Xiaoyong Xu[4], Liyuan Long[1], Jinlei Zhang[1], Jun Dai[3], Junhong Guo[5], Jiancang Shen[1], Shuang Li[2], Lizhe Liu[1] & Xinglong Wu[1]

Photocatalytic hydrogen evolution from water has triggered an intensive search for metal-free semiconducting photocatalysts. However, traditional semiconducting materials suffer from limited hydrogen evolution efficiency owing to low intrinsic electron transfer, rapid recombination of photogenerated carriers, and lack of artificial microstructure. Herein, we report a metal-free half-metallic carbon nitride for highly efficient photocatalytic hydrogen evolution. The introduced half-metallic features not only effectively facilitate carrier transfer but also provide more active sites for hydrogen evolution reaction. The nanosheets incorporated into a micro grid mode resonance structure via in situ pyrolysis of ionic liquid, which show further enhanced photoelectronic coupling and entire solar energy exploitation, boosts the hydrogen evolution rate reach up to 1009 $\mu$mol g$^{-1}$ h$^{-1}$. Our findings propose a strategy for micro-structural regulations of half-metallic carbon nitride material, and meanwhile the fundamentals provide inspirations for the steering of electron transfer and solar energy absorption in electrocatalysis, photoelectrocatalysis, and photovoltaic cells.

[1] Key Laboratory of Modern Acoustics, MOE, Institute of Acoustics and Collaborative Innovation Center of Advanced Microstructures, National Laboratory of Solid State Microstructures, Nanjing University, Nanjing 210093, China. [2] Key Laboratory of Advanced Functional Materials of Nanjing, Nanjing Xiaozhuang University, Nanjing 211171, China. [3] Department of Physics, College of Science, Jiangsu University of Science and Technology, Zhenjiang 212003, China. [4] School of Physics Science and Technology, Yangzhou University, Yangzhou 225002, China. [5] School of Optoelectronic Engineering and Grüenberg Research Centre, Nanjing University of Posts and Telecommunications, Nanjing 210023, China. These authors contributed equally: Gang Zhou, Yun Shan, Youyou Hu, Xiaoyong Xu. Correspondence and requests for materials should be addressed to L.L. (email: lzliu@nju.edu.cn) or to X.W. (email: hkxlwu@nju.edu.cn)

On the journey to pursue renewable green energy to solve the global energy crisis and environmental pollution, hydrogen is regarded as an ideal energy owing to its environmental friendliness and high energy capacity[1,2]. It is of great interest to utilize solar energy to acquire hydrogen from water, especially in light of the environmental cost. However, in many applications, the lack of stable, efficient, and inexpensive catalysts for such systems remains a principal problem. Inspired by rapidly developing exploratory study, the economical and abundant metal-free photocatalysts, such as carbon quantum dots[3,4], black phosphorus[5,6] and graphitic carbon nitrides[4,6–9], have been proposed to replace the metal-contained, especially for noble metal Pt catalyst. Among the above metal-free semiconducting materials, photocatalysts with large number of exposed active sites, low resistance for carrier transfer and wide solar light response have been identified as promising candidates for hydrogen evolution reaction (HER)[10]. In this context, the introduction of metal-free half-metallic feature[11–13] (a material with complete spin polarization at Fermi level: the majority is metallic and the minority is semiconducting) into semiconducting nanosheets can meet the above-mentioned requirements. The half-metallicity, ensuing both the maintenance of quick electron transfer and efficient electron-hole separation owing to introduction of triplet and singlet excited states, is conducive to solar exploitation[14]. More importantly, some new active sites are expected to appear on half-metallic nanostructure surfaces.

Herein, we present a solar $H_2$ production system using the half-metallic carbon nitride [hm-$C(CN)_3$] nanosheets incorporated into an artificial nanotube array as the photocatalyst. The designed various nanostructures are widely used in solar cells to enhance light absorption[15–18]. In our system, this artificial nanotube array acts as not only micro grid mode resonance to tune optoelectronic coupling process for enhanced utilization of solar energy but also the template to control the size of carbon nitride nanosheets. Such a use of the half-metallic carbon nitride nanosheets into a photocatalytic system in combination with the micro grid artificial microstructure leads to efficient HER.

## Results

**Preparation and characterization of half-metallic carbon nitride samples.** The density functional theory (DFT) prediction was implemented and the calculated results show that the layered carbon nitride [$C(CN)_3$] has the expected intrinsic half-metallicity and suitable band gap (2.06–2.33 eV) to harvest more solar energy (Supplementary Fig. 1)[14]. The incorporation of two-dimensional (2D) hm-$C(CN)_3$ nanosheets into the artificial nanotube arrays (nanoporous anodic aluminum oxide membrane, AAM) [named as MG@hm-$C(CN)_3$] to form micro grid mode resonance structure was achieved via in situ pyrolysis of imidazolium-based ionic liquid, and meanwhile the bulk hm-$C(CN)_3$ samples were prepared without AAM for characterization and comparison. Here, the pyrolytic condition for hm-$C(CN)_3$ sample was determined via the scanning thermal gravimetric analysis of the ionic liquid precursor (Supplementary Fig. 2). Detailed description is given in the Methods and Supplementary Fig. 3. Energy dispersive X-ray (EDX) spectra disclose that hm-$C(CN)_3$ bulk sample and nanosheets derived from the ionic liquid precursor have the same C/N ratios (~ 58 at% for C and ~ 42 at% for N). These values are close to 4/3 which is the expected value for hm-$C(CN)_3$ sample.

The experimental and calculated X-ray diffraction (XRD) patterns (Fig. 1a) confirm that the strongest diffraction peak located at ~ 25.75° is attributed to the (002) peak of bulk hm-$C(CN)_3$ and the high angle peak at ~ 43.86° is indexed into (200) peak, respectively[19]. The photo of the hm-$C(CN)_3$ powder in the inset displays a black metallic luster, which is favorable to solar absorption[20]. The bulk hm-$C(CN)_3$ has a sheet-like shape with a planar size of 5–250 μm (Supplementary Fig. 4a). Atomic force microscopy (AFM) image indicates these sheets are of 0.7 ~ 1.1 μm in thickness (Supplementary Fig. 4b). EDX analysis confirms that these hm-$C(CN)_3$ sheets are mainly composed of C and N elements (Supplementary Fig. 5). The corresponding element mapping images in Supplementary Fig. 6 show a homogeneous nature of the as-prepared sheets, in which C and N are uniformly distributed in the mapping image of hm-$C(CN)_3$. XPS further

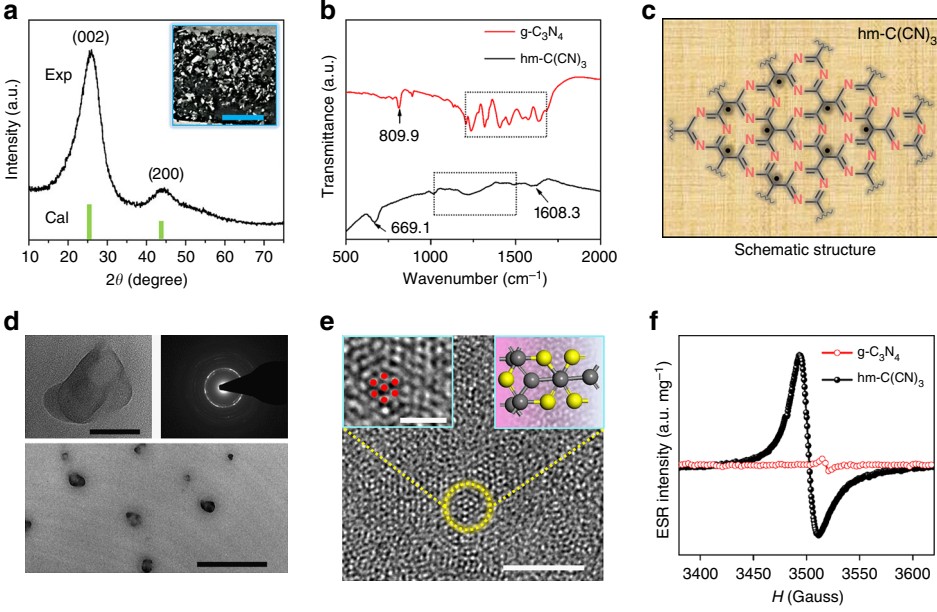

**Fig. 1** Characterization of hm-$C(CN)_3$. **a** XRD patterns of the bulk hm-$C(CN)_3$ materials (Scale bar, 1 cm). **b** FTIR spectra of g-$C_3N_4$ and bulk hm-$C(CN)_3$. **c** Schematic representation of chemical structure for hm-$C(CN)_3$ material. **d** Top panel: applied TEM image (Scale bar, 20 nm) and SAED pattern of R@hm-$C(CN)_3$ nanosheets. Bottom panel: TEM image of R@hm-$C(CN)_3$ nanosheets (Scale bar, 200 nm). **e** HR-TEM image of R@hm-$C(CN)_3$ nanosheets (Scale bar, 2 nm). Left inset: locally amplified HR-TEM image (Scale bar, 1 nm). Right inset: atomic structural model. **f** ESR spectra of g-$C_3N_4$ and hm-$C(CN)_3$ nanosheets

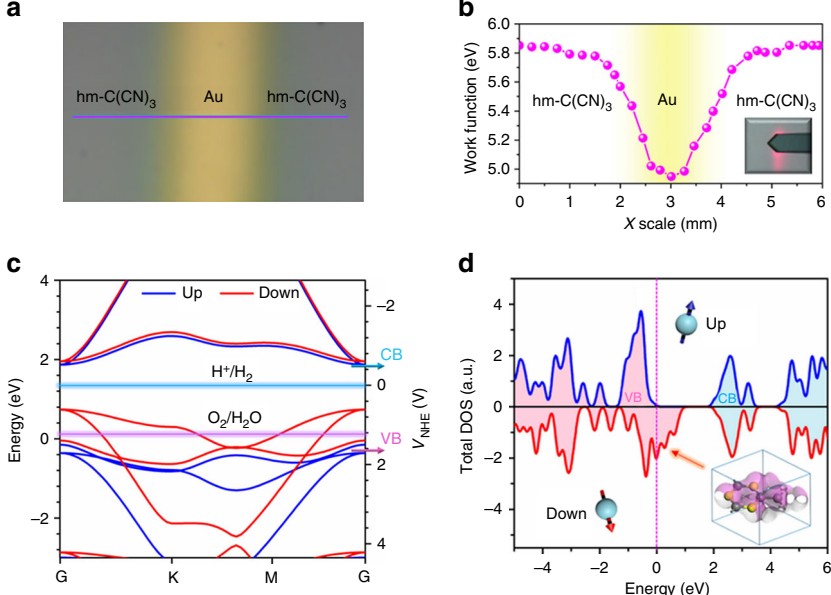

**Fig. 2** Electronic structure of hm-C(CN)$_3$ nanosheets. **a** Photograph of the hm-C(CN)$_3$ nanosheet film patterned with the Au line for the scanning Kelvin probe measurement. **b** Line-scan data show the Fermi level of hm-C(CN)$_3$ at each position on the purple line in **a**. **c** Band structure for hm-C(CN)$_3$. The position of the reduction level for H$^+$ to H$_2$ is indicated by the dashed blue line and the oxidation potential of H$_2$O to O$_2$ is indicated by the purple dashed line just above the valence band. The blue and pink arrows represent the CB and VB positions of hm-C(CN)$_3$ nanosheets which acquired from the experiment. **d** Spin-resolved total density of state for hm-C(CN)$_3$. The short pink dotted line indicates the Fermi level

evidences the element composition and the successful preparation of hm-C(CN)$_3$ (Supplementary Fig. 7). To better understand the structure of hm-C(CN)$_3$, the Fourier transform infrared (FTIR) spectra were provided to compare with g-C$_3$N$_4$. As shown in Fig. 1b, the FTIR spectrum of g-C$_3$N$_4$ shows strong vibration bands at 1200–1650 cm$^{-1}$ (marked by black box) and 809.9 cm$^{-1}$, which are attributed to the stretching vibrations of the CN heterocycles and out of plane bending vibration of the aromatic ring, respectively[9,20,21]. However, the FTIR spectrum of hm-C(CN)$_3$ mainly displays the stretching vibrations of C-(C)$_3$ (669.1 cm$^{-1}$), CN (1000–1500 cm$^{-1}$) (marked by black box) and C=N (1608.3 cm$^{-1}$). Obviously, the difference in vibrational energy is ascribed to their different chemical structures. Based on XRD, XPS analysis and previous report[19], the chemical structure of hm-C(CN)$_3$ can be inferred as shown in Fig. 1c, which is further confirmed by the solid state $^{13}$C nuclear magnetic resonance (NMR) spectra (Supplementary Fig. 8).

The transmission electron microscope (TEM) images indicate that the obtained hm-C(CN)$_3$ nanosheets after facile removal of AAM using 0.5 M NaOH solution emerge with a planar size of 20 ~ 60 nm (bottom panel in Fig. 1d). These nanosheets, without further liquid exfoliation, have much smaller size than that of bulk hm-C(CN)$_3$, demonstrating that the structure-assisted in situ pyrolysis of ionic liquid is a potential strategy to obtain carbon nitride nanosheets. The selected area electron diffraction pattern reveals that faint diffraction rings are assigned to the {002} and {200} with lattice spacing of 0.47 and 0.21 nm (upside in Fig. 1d), indicating the polycrystalline structure of hm-C(CN)$_3$ nanosheets. This is well consistent with the broad diffraction peaks from bulk hm-C(CN)$_3$ in XRD pattern. The high-resolution transmission electron microscope (HR-TEM) image for a typical crystalline region shows the arrangement of atoms (left inset in Fig. 1e), which is accordant with the theoretical simulation displayed in the right inset of Fig. 1e. It is worthwhile pointing out that some disordered structures may exist in our system, leading to more dangling bonds[10,22], which can improve optical absorption (Supplementary Fig. 9). However, this disordered structure makes the protons difficult to bond to the catalyst surface, resulting in the slow HER kinetics (Supplementary Fig. 10). Therefore, we point out that the main contribution to the HER is from active sites at crystalline region of hm-C(CN)$_3$ material. As shown by electron spin resonance (ESR) spectra in Fig. 1f, compared with C$_3$N$_4$ materials[10], the superfluous C element in hm-C(CN)$_3$ introduces larger number of lone pair of electrons out of the aromatic rings (marked by red circle, see atomic model in Supplementary Fig. 7), which can be used as additional active sites. The increased ESR intensity of hm-C(CN)$_3$ corroborates the improved delocalization of electrons and appearance of carbon atoms as adsorption sites.

**Electronic structures of the half-metallic carbon nitride nanosheets**. For highly efficient photocatalytic hydrogen evolution, the relative position of the conduction band minimum (CB) and valence band maximum (VB) and their absolute energies with respect to the reduction and oxidation levels are decisive elements[7]. To determine these parameters of hm-C(CN)$_3$ nanosheets, the work function of the hm-C(CN)$_3$ nanosheets was obtained by scanning Kelvin probe force microscopy, as shown in Fig. 2a, b. The Au line was patterned on the hm-C(CN)$_3$ nanosheets as a reference (Fig. 2a), which has a known work function of 4.9 eV (the valley value in Fig. 2b). The line-scan data (Fig. 2b) show the uniformity of the as-prepared hm-C(CN)$_3$ nanosheets and the average Fermi level ($E_F$) across the middle area is ~ 5.84 eV. The band gap is evaluated as 2.19 eV from the linear potential scans (Supplementary Fig. 11). Taking the ultraviolet photoelectron spectroscopy (UPS) (Supplementary Fig. 12)[4], band gap and Fermi level into consideration, the CB and VB positions of hm-C(CN)$_3$ nanosheets are evaluated as − 0.47 and 1.72 V, respectively, as shown by the blue and pink arrows in Fig. 2c. The state-of-the-art hybrid functional (HSE06) calculations show that band gap of hm-C(CN)$_3$ decreases from 2.33 eV for monolayer to 2.06 eV for multilayer (Supplementary Fig. 1), and the band structure of monolayer is shown in Fig. 2c. The hm-C(CN)$_3$ displays a non-isotropic electronic structure

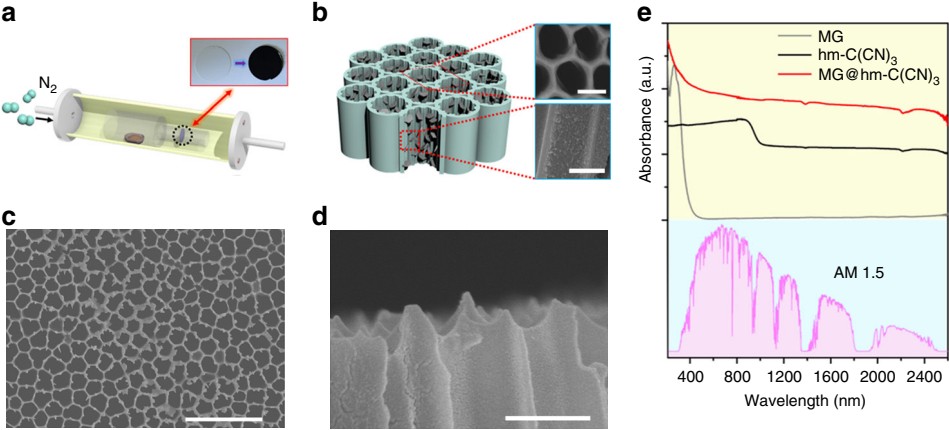

**Fig. 3** Micro grid mode resonance based artificial microstructure. **a** Schematic key steps involved in MG@hm-C(CN)$_3$ synthesis. Inset: digital camera images of a bare nanoporous template (white) sample and a MG@hm-C(CN)$_3$ sample (black). **b** Self-assembly of MG@hm-C(CN)$_3$ nanosheets on nanoporous templates to form artificial microstructure (Scale bar, 150 nm). **c** Top-view (Scale bar, 1 μm) and **d** cross-sectional (Scale bar, 200 nm) SEM images of the MG@hm-C(CN)$_3$. **e** Experimental absorption spectra measured by an integrated sphere in the visible and near-infrared range. MG represents the sample of nanoporous anodic aluminum oxide membrane (AAM), hm-C(CN)$_3$ represents the sample of hm-C(CN)$_3$ bulk material, MG@hm-C(CN)$_3$ represents the sample of hm-C(CN)$_3$ nanosheets incorporated into AAM, respectively

with a direct band gap (blue line: ~ 2.33 eV) at the G point, which is in coincidence with the experimental measurement. Importantly, the spontaneous spin polarization makes spin-down bands (marked by down) obviously across the $E_F$ to display metallic characteristic[14], whereas the spin-up bands (marked by up) are obviously lower than $E_F$ and show semiconducting feature. This interesting half-metallic feature is experimentally observed in the valence band spin-resolved photoemission spectra (PES) near $E_F$ at 300 K of the hm-C(CN)$_3$ sample and confirmed by DFT calculations (Supplementary Figs. 13–18, Supplementary Table 1 and Supplementary Note 7). The spectrum originated from the majority spin is extended to $E_F$ and displays an obvious metallic Fermi cutoff, although that of the minority spin reduces sharply at binding energy of ~ 0.6 eV and the spectral weights disappear around $E_F$, displaying a insulating gap. Furthermore, DFT calculations disclose that the radical C sites, resulting in half-metallic feature, can be easily decorated by hydrogenation than that of defects. (Supplementary Figs. 13–14 and Supplementary Table 1). Simultaneously, the $^{13}$C NMR spectra in Supplementary Fig. 15 indicates that the peak 1 originating from radical C sites slightly shifts after hydrogenation treatment, further confirming that hydrogen deactivation occurs at radical C sites and no additional defect structures are introduced. After hydrogenation treatment, the spin splitting between majority and minority in PES spectra disappear completely, which indicates that spin polarization is mainly originated from radical C sites (Supplementary Figs. 16–17 and Supplementary Note 1). The spectral differences between PES spectra and spin-resolved DOSs (densities of states) stem from lattice distortion or deformation in sample preparation, which can be confirmed by broad XRD pattern and DFT calculations (Supplementary Figs. 18–19).

Besides, the spontaneous spin polarization brings about remarkable ferromagnetic hysteresis as shown in Supplementary Fig. 20. Under optical excitation, the photogenerated carriers occupying different spin states can transform each other. The spin-resolved total density of state in Fig. 2d shows that the hm-C(CN)$_3$ surface is fully filled with metallic spin-down states (see inset), which is more advantageous to carrier transfer. In addition, orbital analysis reveals that the half-metallicity mainly originates from the $p$ orbital of three N atoms at triazine ring[11,14]. The C atom outside of triazine ring (marked by circle in Supplementary Fig. 7) will inject a hole into this system, which

may significantly alter its electronic properties and photocatalytic activity. These results evidence that hm-C(CN)$_3$ is a potential photocatalyst for hydrogen evolution.

**Design of micro grid mode resonance structure for hydrogen evolution.** The hm-C(CN)$_3$ nanosheets were combined with micro grid mode resonance structure for better capture of light. Figure 3a schematically depicts the in situ pyrolysis process. The original transparent nanoporous anodic aluminum oxide membrane turns to light-proof black after carbon nitride incorporation (inset of Fig. 3a), which intuitively implies that our as-prepared artificial microstructure has excellent broadband solar light absorption. The structural diagram and scanning electron microscope images in Fig. 3b–d reveal that the carbon nitride nanosheets tend to be closely packed along the sidewalls of the nanopores of the AAM. Compared with AAM absorption (named as MG), the pristine hm-C(CN)$_3$ nanosheets can absorb the nearly whole solar energy (bottom panel in Fig. 3e) (see measurement details in Supplementary Methods), owing to their special electronic structure (Fig. 2c, d). When the hm-C(CN)$_3$ nanosheets are incorporated into nanopores of the AAM, enhanced entire solar absorption is obtained (Fig. 3e). This absorption enhancement is ascribed to the micro grid mode resonance, which leads to the changes in effective dielectric constant at nanotube wall and the light transmission route.

To explain this optical mechanism, the distribution of light field in the artificial nanotubes is calculated by the finite difference time domain (FDTD) method and shown in Fig. 4a. The hexagonal nanotube arrays with hm-C(CN)$_3$ nanosheets (pink layer) on the inside wall are constructed in the top of Fig. 4a. The FDTD simulations disclose that the distribution of electric field at nanotube surface is enhanced (the bottom of Fig. 4a and Supplementary Fig. 21). It is well known that polarization is an inherent property of light waves[23]. Maxwell's electromagnetic theory holds that light wave is a shear wave, light vector direction of which is always perpendicular to the propagation direction. When the natural light projected at the interface of the media undergoes reflection and refraction, light vector can be decomposed into two parts (Fig. 4b and Supplementary Fig. 22). One part parallel to the incident surface is called the P wave (marked by blue arrows: $E_{1p}$) (see details in the Supplementary Information), and the other part

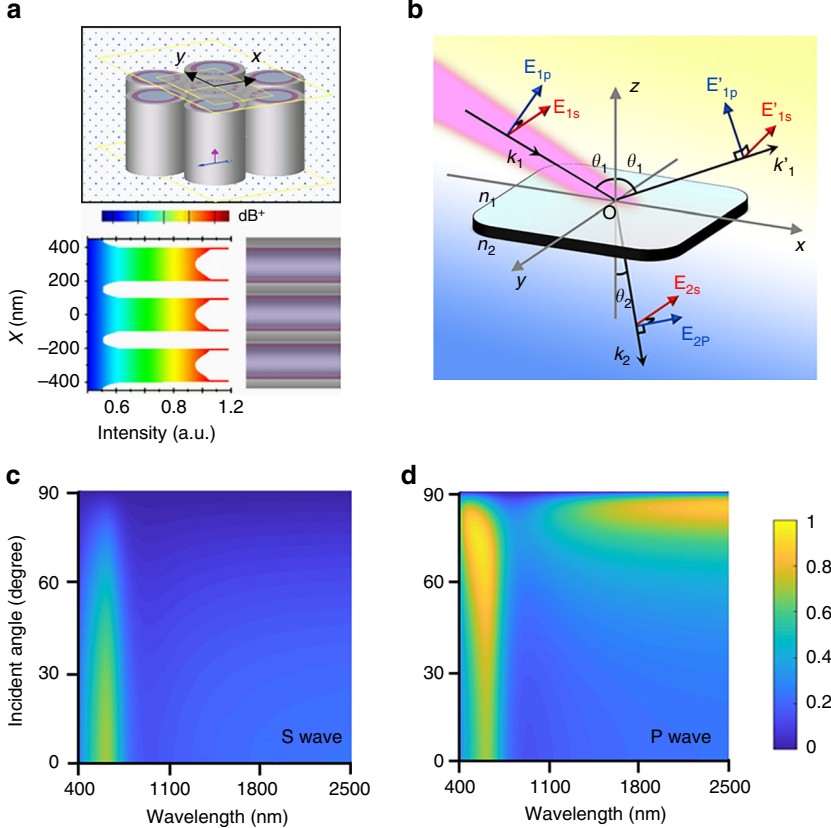

**Fig. 4** Simulation of micro grid mode resonance enhanced absorption based polarization optics. **a** Schematic diagram of the model system: the hexagonal nanotube arrays with deposited hm-C(CN)$_3$ (top) and Cross-section (bottom) of the simulated electric field distribution of MG@hm-C(CN)$_3$. **b** Ep and Es for the two mutually perpendicular components of Electric vector. Absorption zones of **c** S-wave and **d** P wave versus wavelength and incident angle

perpendicular to the incident surface is called the S wave (marked by red arrows: $E_{1s}$). Compared with absorption of S waves in Fig. 4c, the P waves have two strong absorption areas in the large incidence angle (Fig. 4d), which indicates that the P wave behavior is more beneficial to light absorption in packed hm-C(CN)$_3$ layer. In this special columnar micro grid structure, the incident natural lights are all expressed as P wave on all the planes perpendicular to the nanotube inner walls (Supplementary Fig. 23 and Supplementary Note 2). Therefore, more energy will be introduced into the hm-C(CN)$_3$ nanosheets via P wave behavior, leading to higher absorption capacity. Besides, hm-C(CN)$_3$ nanosheets are incorporated into AAM to construct cylindrical resonators (void resonators), of which the inner walls are rough. Theoretical calculations indicate that cylindrical resonators with rough inner wall can reduce reflections and increase effective optical length, which makes incident lights travel with coherent superposition, resulting in a linear near-infrared (NIR) absorption (Supplementary Fig. 23).

**Photocatalytic hydrogen evolution performance**. Figure 5a shows the electrical conductivity of the multilayered hm-C(CN)$_3$ nanosheets reaches a high magnitude ($10^6$), which is advantageous to the migration of photogenerated carriers. To clarify the influence of van der Waals forces in multilayered hm-C(CN)$_3$ nanosheets on its high electrical conductivity, we plotted the electronic wave function distribution across multilayered hm-C(CN)$_3$ nanosheets for the CB in Fig. 5b. Owing to weak interlayer coupling effect in this 2D construction, built-in electric field induced by photogenerated carriers restrict electrons to intralayer so as to promote electron transfer, resulting in high electrical

conductivity and excellent HER performance. The photocatalytic hydrogen production activities of different samples were detected in a mixed 10% (v/v) triethanolamine aqueous solution under a Xe lamp irradiation with an AM 1.5 G filter (Supplementary Movie 1). As shown in Fig. 5c, the pristine g-C$_3$N$_4$ sheets exhibit a low HER activity because of the rapid recombination of electron −hole pairs though they have similar size to our bulk hm-C(CN)$_3$ sample (∼ 200 μm see Supplementary Fig. 24). The bulk hm-C(CN)$_3$ sheets bring about a enormous improvement in the hydrogen production rate (up to 301 μmol g$^{-1}$ h$^{-1}$) because the excellent optical absorption induces more photogenerated carriers and its half-metallic feature facilitates the rapid separation of photogenerated electron−hole pairs. As expected, MG@hm-C(CN)$_3$ sample shows a much higher hydrogen evolution rate of 1009 μmol g$^{-1}$ h$^{-1}$, ∼ 62.8 times higher than that of pristine g-C$_3$N$_4$ sheets. After removal of the nanotubes [named as R@hm-C(CN)$_3$], the photocatalytic activity drops obviously, but still much higher than that of bulk hm-C(CN)$_3$ sheets. This is because the R@hm-C(CN)$_3$ nanosheets of much smaller size (20 ∼ 60 nm, Supplementary Fig. 25) have more active sites than bulk hm-C(CN)$_3$ sample although the capture of light reduces without micro grid structure. In order to clarify the contributions from lights of different wavelengths to the hydrogen evolution, photocatalytic H$_2$ evolution and On/Off photocurrent curves were carried out under different irradiation lights (Supplementary Fig. 26 and Supplementary Note 3). Both the hydrogen evolution rates and the photocurrents decrease obviously in the absence of NIR or ultraviolet (UV) lights, evidencing that the entire solar energy can make contributions to the hydrogen evolution. The calculated solar-to-hydrogen conversion efficiencies for different samples are described in the Supplementary Fig. 27 and Supplementary

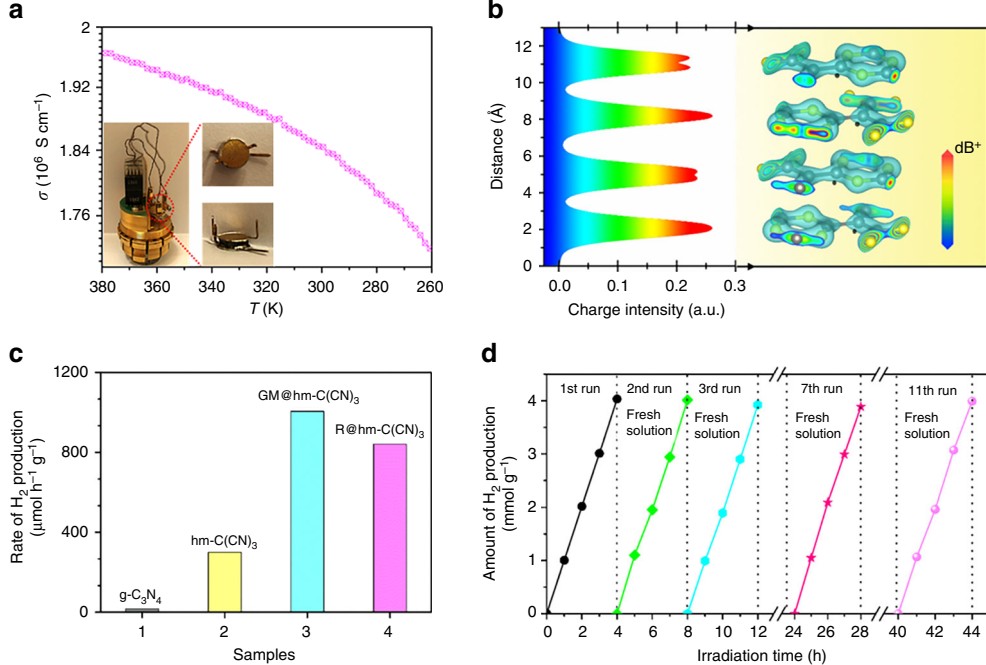

**Fig. 5** Practical application of HER. **a** The electrical conductivity and **b** the charge density distribution and the corresponding multilayered structure of hm-C(CN)$_3$ nanosheets. **c** Comparison of the photocatalytic hydrogen production over different samples under the white light irradiation. **d** Recycling test of photocatalytic hydrogen production over MG@hm-C(CN)$_3$

Methods, and MG@hm-C(CN)$_3$ has a highest value of 2.24 ($\Delta G^* 10^{-6}$) %, which corresponds to an apparent quantum efficiency of 1.03% (see Supplementary Methods for details)[24]. Besides the efficiency of hydrogen evolution, the long-term stability is another important parameter to evaluate the photocatalytic performance. As shown in Fig. 5d, after eleven cycles of repeated trials, MG@hm-C(CN)$_3$ shows negligible change in HER performance, which indicates that MG@hm-C(CN)$_3$ has the excellent photochemical stability. The amount of evolved hydrogen is proportional to the sunlight irradiation time, indicating of the constant hydrogen production under continuous irradiation. More importantly, for practical application, our designed MG@hm-C(CN)$_3$ sample can be easily collected without need of an additional recycle technology, which has obvious advantages over conventional photocatalyst nanomaterials.

**Photocatalytic hydrogen evolution mechanism**. To elucidate the HER mechanism, the potential energy profile is calculated to estimate the adsorption of H$^+$ on the different sites of hm-C(CN)$_3$ nanosheets. The results show that the favorite adsorption sites are located in the neighboring region of C or N atom, especially for C1 and C2 positions (Fig. 6a). When these active sites are covered by H* (atom), the corresponding free energy for hydrogen adsorption is calculated and shown in Fig. 6b. It is generally known that a material displays a good catalytic activity when the free energy of adsorbed hydrogen tends to thermoneutral, i.e., $\Delta G_H \approx 0$. If hydrogen cannot efficiently adsorbs onto catalyst or forms a strong chemical bond, the hydrogen release and proton/electron transfer process also be limited, leading to lower catalytic activity[2,25–27]. DFT calculation indicates that the C2 sites successively introduce nonequilibrium stoichiometry to allow for favorable hydrogen adsorption on this active site as follows: H$^+$ + e$^-$ + * → H*, where * denotes a binding site. After spontaneous hydrogen adsorption occurred at C1 and C2 sites (Fig. 6b), the $\Delta G_H$ at the C2 sites becomes almost thermoneutral, reaching an optimal $\Delta G_H$. This makes the release of molecular hydrogen easier via reactions 2 H* → H$_2$↑ + 2*. In actual HER

process, H$^+$ can unavoidably adsorb onto C4 and C6 sites, but this adsorption cannot lead to hydrogen generation. This is because if the H* adsorbed onto the most active termination C2 site intends to generate molecular hydrogen more efficiently, it has to bond to neighboring H* at C1 site. Thus, the influence of C4 and C6 terminations on HER can be neglected. To further understand the process of hydrogen production, the climbing nudged elastic band method is adopted to study the adsorption, activation, and reaction processes of H$^+$ on the hm-C(CN)$_3$ nanosheets, and the results are shown in Fig. 6c. Generally, the spin triplet state energy (marked by black T line) is smaller than that of spin singlet state (marked by red S line), and triplet-to-singlet conversion happens between S6 and S7. Naturally, the HER process will follow lower energy route (marked by a black arrow). First, a single H atom appears away from site C1 at a distance of 4.903 Å (S1), then spontaneously adsorbs physically onto the site C1 after energy release of ~ 1.72 eV (S3) before overcoming the negligible barrier at S2. Subsequently, another H atom closes (S4) and adsorbs onto the C2 site, then two metastable H* atom configuration forms with a distance of 1.865 Å (S5). Upon photoexcited perturbation, the two adjacent H* atoms begin orbital hybridization and bridge a transition state at a H–H distance of 1.864 Å (S6). To complete this process, activation energy of 0.53 eV relative to the energy difference between the potential energies of two metastable states is required. With phonon assistance, a molecular hydrogen with diagonal bridge configuration is dissociated from the hm-C(CN)$_3$ surface. In this process, the degeneracy of its $\pi^*_{2P}$ orbitals is lifted owing to the hybridization with the C1 site lone pairs, and the singlet state becomes more stable than the triplet (between S6 and S7)[28]. Therefore, for the generation of H$_2$, there must be at least one point in configuration space where the triplet and singlet potential energy surfaces cross (S7), and triplet-to-singlet conversion likely takes place nonradiatively. Eventually, the H$_2$ molecule is released from the hm-C(CN)$_3$ surface with an energy gain of 1.24 eV (S8). The triplet-to-singlet conversion can be calculated as: $P_{ts} = 2[1 - \exp(-V^2/h\nu|Ft - Fs|)]$[29], here $h$, $V$, Fs, and Ft

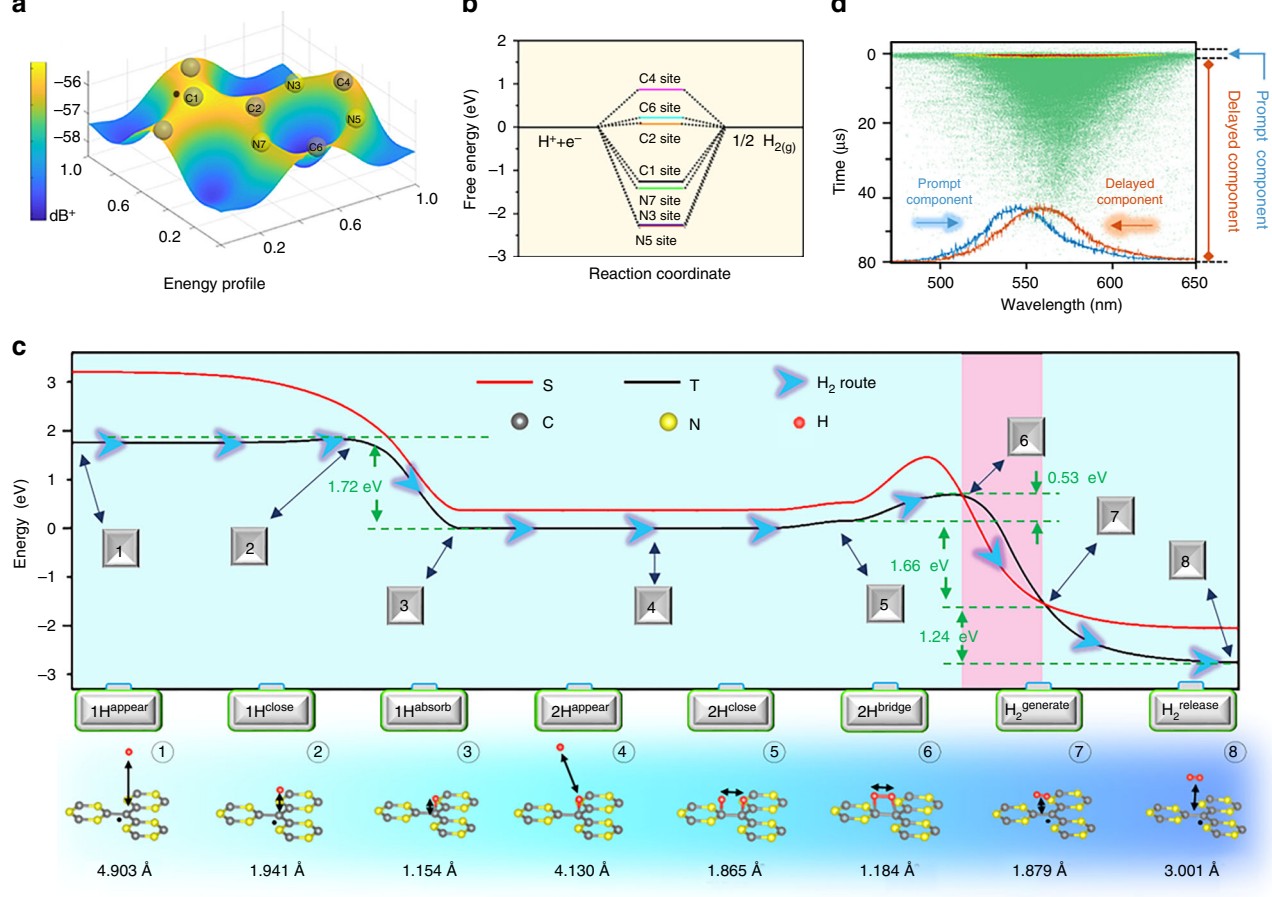

**Fig. 6** Schematic configuration-coordinate diagrams for HER mechanism. **a** The potential energy profile for the adsorption of H$^+$ on the different sites. **b** The process of hydrogen production. **c** Free energy versus the reaction coordinates of HER for different active sites. The singlet (red), triplet (black), and optimal process (marked by a black arrow) as function of the H atom distance are displayed for comparison. The accompanying atom configures are shown in lower plane. **d** Streak image and photoluminescence spectra of R@hm-C(CN)$_3$ nanosheets at 298 K

are Planck's constant, the spin-orbit matrix element, slopes at energy crossing point between triplet and singlet states, respectively. By using $V = 122$ cm$^{-1}$ for describing H$_2$ dynamic process, we find Pts = 0.78 at 300 K[28], which confirms that the triplet-to-singlet conversion plays an irreplaceable role in this HER mechanism.

To experimentally confirm the singlet-to-triplet conversion, the streak image and the time-resolved photoluminescence (PL) spectra are shown in Fig. 6d. The transient photoluminescence spectrum can be resolved into the prompt emissions ($t < 50$ ns) and the long tail components ($t > 50$ ns). The prompt emissions mainly come from singlet excited state, and long tail components are originated from the intersystem crossing from singlet-to triplet excited states. The obvious red-shifts in two time-dependent PL spectra disclose that the triplet excited states have been generated successfully via intersystem crossing process. (See the photoluminescence quantum efficiencies in Supplementary Fig. 28, Supplementary Table 2 and Supplementary Note 8)[30,31]. Therefore, singlet-triplet conversion can be confirmed by this typical difference in transient photoluminescence characteristics. This conversion can also be systematically confirmed by photoluminescence spectra (Supplementary Fig. 29), photoluminescence decay curves (Supplementary Fig. 30) and time-resolved diffuse reflectance spectroscopy (Supplementary Figs. 31–32 and Supplementary Notes 4–5). In addition, the long-lived components of transient photoluminescence and half-field light-induced electron spin resonance spectra are markedly depressed after

hydrogenation or oxygen solution, which confirms that the generation of excited triplet states is limited strongly (Supplementary Fig. 33 and Supplementary Note 6). It is well known that increasing carrier separation and transfer plays a critical role in improving HER performance. In our R@hm-C(CN)$_3$ materials, the carrier recombination lifetime is extended from ns to μs via singlet-triplet conversion (Supplementary Table 2, and Supplementary Figs. 29–33), greatly increasing the probability of electrons transferring to active sites to participate in HER. In addition, the generation of excited triplet states also can be confirmed by half-field light-induced electron spin resonance spectra (Supplementary Fig. 33). This strategy allows for manipulation of electron-hole recombination and can be generalized to other photocatalytic materials to enhance HER performance.

## Discussion

In conclusion, we have demonstrated that half-metallic C(CN)$_3$ nanosheets can be facilely synthesized and incorporated into designed micro grid mode resonance systems via in situ pyrolysis of ionic liquid to tune optical transmission for highly efficient photocatalytic hydrogen evolution. The artificial microstructure not only sharply enhances solar exploitation but also obviously improves photocatalyst's stability and recyclability. More importantly, this microstructure makes it possible to prepare size-controllable carbon nitride materials without liquid exfoliation. In view of the great variety of microstructures and ionic liquids,

other materials of special features are foreseen in the areas of photocatalysis, electrocatalysis, and photovoltaic cells.

## Methods

**Synthesis of MG@hm-C(CN)₃ micro grid resonance structures.** All chemicals were of analytical grade and used as received without further purification. First, 0.01 mol 1-butyl-3-methylimidazolium chloride and 0.01 mol potassium tricyanomethanide (toxic by inhalation, in contact with skin and if swallowed) were dissolved in 30 ml deionized water. Then, the mixture was stirred under the nitrogen current for 30 min at room temperature to ensure complete reaction (step 1). Subsequently, the reaction solution was evaporated in 80 °C water bath in vacuum to remove the solvent. The residual was dissolved in 10 ml absolute ethanol followed by centrifugation to remove the precipitant. The supernatant was evaporated again. The residual was purified with absolute ethanol following the same procedure for several times. The obtained claybank liquid was dried under 60 °C for 48 h in a vacuum environment oven and as the precursor (step 2). The second pyrolysis process was employed to anchor hm-C(CN)₃ nanosheets onto the surface of the nanopores of template, as illustrated schematically in Fig. 3a. Typically, 0.5 g oily precursor as a source material was placed in a ceramic boat, and then was inserted into the middle of the small quartz tube inside a horizontal tube furnace. A cleaned original transparent nanoporous AAM template was put vertically downstream of the oily precursor, at a distance of 15 cm. Before heating, the system was purged with 517 sccm (standard cubic centimeters per minute) high-purity nitrogen (N₂ 99.999%) for 30 min. Then, the pressure was reduced to $7.5 \times 10^{-2}$ Torr for the duration of the reaction. After that, the furnace was heated, with a heating rate of 5 °C min⁻¹, to a temperature of 650 °C. It was kept at this temperature for 2 h with an N₂ flow of 526 sccm (step 3). After the system was cooled down to room temperature, a metalescent product was found deposited onto the AAM. The pyrolytic temperature for bulk hm-C(CN)₃ sample under N₂ carrier gas was determined via TGA (Supplementary Fig. 2). The TGA profile discloses that heating the ionic liquids at 300 °C results in the formation of partially solidified liquid–solid intermediate and heating in the range of 450–550 °C gives solid products. Polymerization of [C(CN)₃]⁻ in the temperature range of 300–400 °C may follow a similar dynamic cyclotrimerization reaction to the condensation of aromatic nitriles, cyanamide, and acetylenes, which is accompanied by the decomposition of the corresponding ionic liquids cations. We prepared the bulk hm-C(CN)₃ sample with the temperature maintained at ~ 500 °C. In preparation of MG@hm-C(CN)₃ samples, the AAM template was put vertically downstream of the oily precursor, and the annealing temperature was increased to 650 °C to optimize the formation of hm-C(CN)₃. After considering a series of distance (5, 10, 15, 20 cm) from the reaction precursor, we found that the 15 cm was the optimal separation. The preparation route is schematically displayed in Supplementary Fig. 3.

**Characterization.** The morphology and microstructure were characterized by field-emission scanning electron microscopy (Hitachi, S4800), HR-TEM (JEOL-2100) equipped with X-ray energy dispersive spectrum, X-ray photoelectron spectroscopy (XPS, PHI5000 VersaProbe), Tapping-mode AFM (Nanoscope IIIA) and X-ray diffractometer (XRD-7000, Shimadzu) with Cu $K_\alpha$ radiation ($\lambda =$ 0.15406 nm). The UV-vis absorption spectra were obtained by the diffuse reflection method on a spectrophotometer (Varian Cary 5000) in the range from 200–2600 nm equipped with an integrated sphere attachment and with BaSO₄ as a reference. The FTIR spectrum was recorded on a FTIR spectrometer (Spectrum One, Perkin Elmer) using a standard KBr pellet technique. UPS measurements were performed with an unfiltered HeI (21.22 eV) gas discharge lamp at a total instrumental energy resolution of 100 meV. Raman spectra were recorded on a Raman microscope (NR-1800, JASCO) using a 514.5 nm argon ion laser. Solid state NMR for 13C magic angle spinning (MAS) measurements were carried out on a Bruker Avance III (600 MHz) spectrometer using a standard Bruker 4 mm double-resonance H-X MAS probe. The femtosecond diffuse reflectance spectra were measured by the pump and probe method using a regeneratively amplified titanium sapphire laser (Spectra-Physics, Spitfire Pro F, 1 kHz) pumped by a Nd:YLF laser. PL measurements were performed using an Edinburgh FLS-980 PL spectrometer. No filter was used for PL measurements. For the PL lifetime measurements, a 375 nm picosecond pulse laser (0.5 mW) was used as the excitation source. The room temperature magnetic properties of samples were measured using a superconducting quantum interference device.

**Photocatalytic hydrogen evolution.** The photocatalytic hydrogen evolution activities of different samples were tested by using a photocatalytic activity evaluation system (Labsolar-III(AG), Perfectlight, Beijing). The light source was a 300 W Xe lamp including an AM 1.5 G filter. The Xe lamp was placed 10 cm away from the reactor. The focused intensity ($I$) on the reactor was adjusted to 100 mWcm⁻². Various photocatalysts with the same quality were directly dispersed into 100 ml of the 10% (v/v) triethanolamine mixture solution. Before irradiation, the whole system was sealed and vacuumed by a mechanical pump to eliminate any gas impurities. Then, the produced hydrogen was extracted every 1 hour with an on-line gas chromatograph (9790 II, Fuli, Zhejiang) equipped with a TCD detector and Ar gas carrier. The generated amount of hydrogen was evaluated according to the

fitted standard curve. During the photocatalytic process, the fluid cooling water was used to guarantee the reactant solution at room temperature.

**Electric near-field simulation details.** The electromagnetic simulation is performed using a Finite-Element Method (FEM). In addition, the refractive index of water is set to be dispersive. All the physical domains are surrounded by perfectly matched layers to absorb all the outgoing waves, thus minimizing any reflection. Plane waves are used as excitation sources in all simulations along the Z axis. For better clarification, the intensity of electrical field distribution is visualized in a square scale.

Maxwell's equations are rigorously solved by three-dimensional finite-element method. The incident, internal, and scattered electromagnetic fields of nanosheet can be described by introducing a series of cuboid harmonics. The absorption coefficient of the light wave is related to the polarization, wavelength, and incident angle of the light. According to the Fresnel formula, the relationship between the absorption coefficient of the medium for TE polarization (s wave) and TM polarization (p wave) and the wavelength and incident angle can be calculated separately.

$$R_{\mathrm{s}} = \frac{\sin^2(\theta_1 - \theta_2)}{\sin^2(\theta_1 + \theta_2)}, \tag{1}$$

$$T_{\mathrm{s}} = \frac{n_2 \cos\theta_2}{n_1 \cos\theta_1} \cdot \frac{4 \sin^2\theta_2 \cos^2\theta_1}{\sin^2(\theta_1 + \theta_2)}, \tag{2}$$

$$R_{\mathrm{p}} = \frac{\tan^2(\theta_1 - \theta_2)}{\tan^2(\theta_1 + \theta_2)}, \tag{3}$$

$$T_{\mathrm{p}} = \frac{n_2 \cos\theta_2}{n_1 \cos\theta_1} \cdot \frac{4 \sin^2\theta_2 \cos^2\theta_1}{\sin^2(\theta_1 + \theta_2)\cos^2(\theta_1 - \theta_2)}. \tag{4}$$

$R_{\mathrm{s}}$, $T_{\mathrm{s}}$, $R_{\mathrm{p}}$, and $T_{\mathrm{p}}$ are, respectively, the reflection and transmission coefficients of s wave and p wave. $n_1$ and $n_2$ are the refractive index of the incident and outgoing regions. $\theta_1$ and $\theta_2$, the incident angle and refraction angle satisfy the law of refraction.

**Theoretical calculation.** The calculations are performed by using plane-wave basis Vienna ab inito simulation package code with generalized gradient approximation[32]. The cutoff energy for expanding Kohn-Sham wave functions is chosen to be 460 eV and the vacumm space of 20 Å is used to avoid the interaction between periodical images, respectively. The Monkhorst-Pack $k$-points grid is $8 \times 8 \times 1$ and all forces on the free ions is 0.03 eV/Å, which have been tested to be well converged. The state-of-the-art hybrid functional calculations based on the Heyd-Scuseria-Ernzerhof (HSE06) functional have been adopted to calculate the spin polarization and half-metallicity in a primitive carbon nitride unit cell. The activation energies are calcuated using the climbing nudged elastic band method[33]. Electromagnetic simulations are obtained by a FEM.

**Data availability.** The data that support the findings of this study are available from the corresponding authors on request.

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

## Acknowledgements

This work was supported by National Basic Research Programs of China under Grants Nos. 2017YFA0303200, 2018YFA0306004, and 2014CB339800, National Natural Science Foundation (Nos. 11404162 and 11674163), and Natural Science Foundation of Jiangsu Province (BK20171332 and BK20161117). This work was also supported by the Fundamental Research Funds for the Central Universities (0204-14380066 and 0204-14380083) and high Performance Computing Centers of Nanjing University and Shenzhen. Partial support was from the Postgraduate Research & Practice Innovation Program of Jiangsu Province (KYCX17_0036).

## Author contributions

G.Z performed the experiments and co-wrote this manuscript. Y.S and X.Y.X prepared the samples and proofread the manuscript. J.D performed the theoretical calculation and co-wrote the manuscript. J.C.S plotted the figures. J.L.Z and J.H.G designed the experimental setup. L.Y.L and S.L provided the calculation resources. Y.Y.H co-wrote the manuscript. L.Z.L and X.L.W analyzed the data and wrote the manuscript.

## Additional information

**Competing interests:** The authors declare no competing interests.

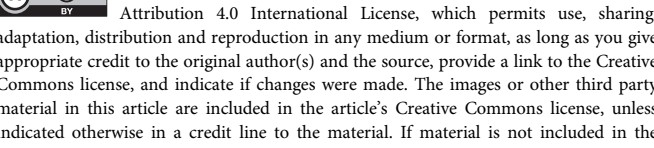

