## [Peer Review File · Nature Communications]

Reviewers' comments:

Reviewer #1 (Remarks to the Author):

1) Summary

The authors report their work toward carbon nitride nanosheets with microgrid resonance structures for improved photocatalytic hydrogen evolution. The material is initially made through a solution chemistry process and is suggested to exhibit "half-metallic" properties based on its broad spectral absorption. The authors report that this material can then be heated in vacuum to become attached to a microstructured membrane comprising hexagonal nanotube arrays. The authors report that this structure further broadens the spectral absorption. The authors claim that the material in their purported microstructure shows a 62x catalytic improvement over pristine graphitic carbon nitride. The authors attribute this enhancement to increased light absorption and rapid separation of photogenerated excitons. To elucidate the photochemical mechanism, the authors calculated potential energy surfaces for different H⁺ binding sites of the material and found that the carbon sites appear to facilitate thermodynamically favorable reduction to hydrogen. The authors suggest a mechanism by which triplet-singlet energy crossing occurs during this process where the singlet becomes lower in energy than the triplet. The authors also suggest that this triplet-singlet conversion plays a role in the hydrogen evolution efficiency that they observed.

2. Subject matter of this manuscript suitable for publication in this journal?

The subject matter of the manuscript could be suitable for the journal as it presents a material not previously shown to have photocatalytic activity and shows a substantial increase in hydrogen evolution beyond that of the widely studied carbon nitride. While the authors certainly performed many experiments, the manuscript lacks a cohesive story and several of the claims do not appear to be entirely supported by the data. As a result, I cannot recommend this manuscript for publication in its current form. The work is also difficult to understand due in part due to numerous typographical errors and confusing sentences, but is also not organized in a way that is easy to follow.

3. Do these findings represent a significant advance in the field?

The dramatic increase in hydrogen production rate is certainly significant. However, the characterization of synthetic explanation of the new material is lacking. Using nanostructures to increase light absorption is not new, as this has been employed in solar cell technologies and studied fundamentally previously. If the authors believe that this is the first time that this approach has been used in photocatalysis, this point needs to be emphasized in the paper and supported with sufficient background. The mechanistic insight could be impactful, but the assertion that a singlet-triplet conversion plays a major role in hydrogen production needs further support. Additionally, a comment on if/how this finding could be extended beyond their specific material system would be useful.

4. How would you characterize the overall quality and completeness of the work?

The quality of writing needs significant work, which makes determining the quality of the rest of the work harder to assess. Overall, it seems to be a very interesting material with several good experiments to highlight this (hydrogen evolution, ESR, DFT calculations, and absorption measurements), but several pieces feel unfinished. There needs to be a figure with the material, not just the repeating unit of 3 triazine rings. It is unclear to the readers exactly how those are connected. Additionally, either the synthesis must be cited if it has been performed before, or if a new synthetic procedure, a proposed synthetic route should be provided in the SI. It also must be clarified if the central carbon is a radical. If so, that must be drawn, if not, then a hydrogen must be shown in the Figure 4a and Figure 5a and b. A paper predicting half-metallicity for the radical polymer does predict half-metal character, but is not cited in this manuscript.¹ The manuscript seems lacking in references to prior work, especially with regard to half-metallic materials and previous uses of microstructures for light absorption. Additionally, the manuscript is lacking in experimental support for

the proposed triplet-singlet conversion. Overall, the manuscript lacks a clear message as the importance of half-metallicity in separating photogenerated excitons is highlighted at the beginning, and the importance of triplet excitons is emphasized at the end. The reader is left confused as to if and how these two ideas are connected, if at all.

5. Are the conclusions adequately supported by the data?

To properly characterize the materials presented, more data would be helpful. The material is synthesized through solution chemistry, dissolved in ethanol and concentrated, and then heated at 650°C to adhere to the microstructured template. Since carbon nitride is made from a variety of carbon- and nitrogen-rich materials at this temperature, it seems plausible that this last heating step could be altering the structure to the graphitic carbon nitride form. However, authors only present FTIR and ¹³C NMR of the solution processed material prior to heating. This is also when a full picture of the material would be helpful, because the structure drawn on the ¹³C NMR should have 3 carbon peaks, not 2. Although TEM and diffraction data is presented for both pre- and post-heated material. It would be interesting to form carbon nitride in the micro grid structure and compare the hydrogen efficiencies to confirm there is a difference.

It appears that the half-metallic character of the material is supported by ESR measurements, the magnetic hysteresis shown in the SI, and DFT calculations. But it would be useful for the reader if given some landmarks for typical hysteresis values.

The improvement in hydrogen evolution data seems well-supported. Although the claim that it has "excellent photochemical stability" after only 12 hours of monitoring doesn't seem completely justifiable.

The mechanistic study presented is intriguing, it seems incomplete. The hydrogen adsorption calculation is definitely interesting and helpful to identify the likely catalytic sites. But it is somewhat questionable if the unit analyzed is actually large enough to be representative of the material. It would be helpful if this was further discussed, specifically if the terminations on C4 and C6 have any effect, since in this case they seem to be radicals. Experimentally, the time-resolved data is incomplete for confirming the triplet-singlet conversion as a catalytically meaningful pathway. The PL lifetime presented is certainly suggestive of a triplet, but without providing a PL quantum yield, it is hard to assess what portion of excitations results in triplet formation. Further evidence would be needed to show that it is an energy crossing of the singlet and triplet levels as the reaction proceeds and not some other process such as thermally activated delayed fluorescence. If a binding event is indeed necessary to break symmetry, then performing experiments in aprotic solvents would be useful to confirm. In general, obtaining a triplet energy experimentally would be helpful in drawing an energy landscape diagram to highlight the possible photophysical processes at play. The authors need to explain the origin of their assignment of the positive feature in the time-resolved absorption measurement as a trapped electron. Where does this assignment come from? The long-lived species could also be triplet absorption. The importance of this data is not clearly addressed and is not tied directly to hydrogen evolution. Overall, a remaining question is the relative importance of triplet excitons versus separated charge carriers in the production of hydrogen, as it was previously stated in the manuscript that the half-metallic character led to efficient separation charge carriers.

A small piece but something confusing is the calculations of crystalline versus amorphous, which is mostly in the SI. The chemical structure used for amorphous is no longer the material, it has broken bonds at random, so it is unclear what the significance is of those calculations. At very least it warrants more discussion.

6. Comments on originality and significance

This material appears to be novel, however, it is unclear how original or significant the incorporation into the micro grid structure is for photocatalysis. While the mechanistic study has the potential to be significant, it is incomplete and still unclear.

7. Comments on the quality, clarity, and conciseness of the writing.

The overall quality of the writing is poor. There are several typographical errors and many confusing sections. The manuscript could definitely benefit from further organization and focus on a couple key takeaway messages.

8. Are the figures, tables, etc. superior quality and appropriateness? What needs to be changed?

The figure captions are not sufficiently informative. Even if markings are explained in the text, they should also be in the figure captions to help the reader. Figures 1 and 3 have too many components in them (9 and 11 parts respectively). Especially when formatted into an article, these will likely be too small, the chemical structures shown in 1f are already small and blurry. Several of these images are more appropriate for the SI. Similarly, Figure 2 b and c seems redundant. The caption for Figure 4a refers to the "crystal structure" of the material, but this seems to just be an optimized geometry for stacked pieces of the material. The pieces of Figure 4a seem redundant, it would be simpler to just include one (probably the lower one is more clear). Figure 4b is a bit overwhelming and hard to work through without a detailed figure caption. The arrows on 4b are very blurry.

9. Is satisfactory nomenclature used?

I'm unsure whether "Mirco Grid Mode Resonance" should be capitalized. The nomenclature to refer to the templated material as "N-C(CN)₃-" seems unwise since it is unclear why that letter is chosen and it implies nitrogen to many readers. It is also strange that the "hm" for half metallic was dropped, since presumably, it is the same material.

10. Are potential hazards adequately described?

No hazards stated. Safety concerns for potassium tricyanomethanide should be stated as it has significant health and safety concerns (MSDS says fatal if inhaled or contacted with skin).

11. Are appropriate references provided? Are they correct?

As mentioned above, the manuscript is lacking references for nanostructures used for increased light absorption. It would also be nice to have more background for half-metallic materials addressing other half-metallic materials that have been made and how they are characterized. A paper predicting half-metallicity for the radical polymer predicts half-metal character, but is not cited in this manuscript [Lee, E. C. et al. Chemistry – A European Journal 16, 12141-12146, doi:10.1002/chem.201000858 (2010)]. References to other systems where a crossing of the triplet and singlet energy occurs would also be helpful to support this proposed mechanism, for example, some of the early work of Turro on biradical systems might be helpful here.

12. Suggested Improvements

Improve the clarity of the writing and proof-read more carefully.

See the section on supporting conclusions with data.

Add safety concerns.

Refine figures and provide detailed figure captions.

Reviewer #2 (Remarks to the Author):

A new metal-free carbon nitride with half-metallic characteristic [hm-C(CN)₃] were incorporated into artificial nanotube array through in situ pyrolysis of imidazolium-based ionic liquid, which showed highly photocatalytic activity for hydrogen evolution. But there some questions need you answer.

1. In Figure S13, the author reported that the N-C(CN)₃ sample had a highest solar-to-hydrogen value of 2.24%. Please provide the video with the hydrogen evolution process as a solid evidence and also revise the picture with clear mark.
2. In Figure 4b, the author mentioned about the sample "R-C(CN)₃". However, there is no description of this sample in the manuscript or supporting information.
3. In the reviewer's opinion, the bandgap of hm-C(CN)₃ measured from UV-DRS spectra in Fig. 3e and Fig. S7 cannot reflect the real physical information of this composite semiconductors because of the flat absorption behavior. For example, in a more simplified mode, the reviewer simulated a UV-DRS spectra with a constant absorption coefficient of 1.2 from 200 to 1000 nm as shown in Fig. 1a. Through K-M conversion, Fig. 1b shows a flat bent curve similar to the result in the manuscript. However, the calculated bandgap can be varied significantly depending on the starting point of the tangent and its slope. In fact, all black materials with a flat absorption behavior in visible light region will exhibit similar trend. The authors should check the correctness of the results and give some explanations.

A new metal-free carbon nitride with half-metallic characteristic [hm-C(CN)₃] were incorporated into artificial nanotube array through in situ pyrolysis of imidazolium-based ionic liquid, which showed highly photocatalytic activity for hydrogen evolution. But there some questions need you answer.

1. In Figure S13, the author reported that the N-C(CN)₃ sample had a highest solar-to-hydrogen value of 2.24%. Please provide the video with the hydrogen evolution process as a solid evidence and also revise the picture with clear mark.
2. In Figure 4b, the author mentioned about the sample “R-C(CN)₃”. However, there is no description of this sample in the manuscript or supporting information.
3. In the reviewer’s opinion, the bandgap of hm-C(CN)₃ measured from UV-DRS spectra in Fig. 3e and Fig. S7 cannot reflect the real physical information of this composite semiconductors because of the flat absorption behavior. For example, in a more simplified mode, the reviewer simulated a UV-DRS spectra with a constant absorption coefficient of 1.2 from 200 to 1000 nm as shown in Fig. 1a. Through K-M conversion, Fig. 1b shows a flat bent curve similar to the result in the manuscript. However, the calculated bandgap can be varied significantly depending on the starting point of the tangent and its slope. In fact, all black materials with a flat absorption behavior in visible light region will exhibit similar trend. The authors should check the correctness of the results and give some explanations.

Figure 1. Simulated absorption spectra (a) and its K-M conversion result (b)

Reviewer #3 (Remarks to the Author):

The work by Zhou et al present combined theoretical and experimental studies of carbon nitride nanosheets for photocatalytic hydrogen evolution. The topic is quite interesting, but graphitic carbon nitride has been well studied for solar driven water splitting. Generally photocatalytic water splitting should semiconducting with a band gap over 1.23 eV and has favorable band position for oxygen and hydrogen evolution reaction. The current work reported a metallic carbon nitride. So I feel this could be a serious problem. The metallic C₃N₄ may be only good for electrocatalysis and the claim for photocatalysis and photovoltaics is questionable. In addition, more evidence on the C₄N₃ like structure and experimental characterization of hydrogen production are also needed. Based on these, I have to recommend the rejection for publishing in high impact Journal Nature Communications.

"Half-Metallic Carbon Nitride Nanosheets with Micro Grid Mode Resonance Structure for Highly Efficient Photocatalytic Hydrogen Evolution"

ID: NCOMMS-17-30419A

Response to the report of the reviewer 1

Comment 1. *Summary*

The authors report their work toward carbon nitride nanosheets with microgrid resonance structures for improved photocatalytic hydrogen evolution. The material is initially made through a solution chemistry process and is suggested to exhibit “half-metallic” properties based on its broad spectral absorption. The authors report that this material can then be heated in vacuum to become attached to a microstructured membrane comprising hexagonal nanotube arrays. The authors report that this structure further broadens the spectral absorption. The authors claim that the material in their purported microstructure shows a 62x catalytic improvement over pristine graphitic carbon nitride. The authors attribute this enhancement to increased light absorption and rapid separation of photogenerated excitons. To elucidate the photochemical mechanism, the authors calculated potential energy surfaces for different H^+ binding sites of the material and found that the carbon sites appear to facilitate thermodynamically favorable reduction to hydrogen. The authors suggest a mechanism by which triplet-singlet energy crossing occurs during this process where the singlet becomes lower in energy than the triplet. The authors also suggest that this triplet-singlet conversion plays a role in the hydrogen evolution efficiency that they observed.

Answer: We deeply appreciate the appropriate comments of the reviewer on our work.

Comment 2. *Subject matter of this manuscript suitable for publication in this journal?*

The subject matter of the manuscript could be suitable for the journal as it presents a material not previously shown to have photocatalytic activity and shows a substantial increase in hydrogen evolution beyond that of the widely studied carbon nitride. While the authors certainly performed many experiments, the manuscript lacks a cohesive story and several of the claims do not appear to be entirely supported by the data. As a result, I cannot recommend this manuscript for publication in its current form. The work is also difficult to understand due in part due to numerous typographical errors and confusing sentences, but is also not organized in a way that is easy to follow.

Answer: The manuscript has been improved to reach a good coherence and some supplementary experimental results are provided to confirm our original claims. In addition, some typographical errors and confusing sentences have been revised. Those revised parts are highlighted by red.

Comment 3. *Do these findings represent a significant advance in the field?*

Reply: This question can be considered from three parts.

(1) The dramatic increase in hydrogen production rate is certainly significant. However, the characterization of synthetic explanation of the new material is lacking.

Answer: Thanks for the reviewer's suggestion. In order to explain the preparation of the new material more clearly, the preparation route was schematically described in Figure R1.

Figure R1. The schematic representation of preparation route for hm-C(CN)₃ sample.

Firstly, 0.01 mol 1-butyl-3-methylimidazolium chloride and 0.01 mol potassium tricyanomethanide were dissolved in 30 mL deionized water. Then, the mixture was stirred under the nitrogen current for 30 min at room temperature to ensure complete reaction (step 1). Subsequently, the reaction solution was evaporated in 80 °C water

bath in vacuum to remove the solvent. The residual was dissolved in 10 mL absolute ethanol followed by centrifugation to remove the precipitant. The supernatant was evaporated again. The residual was again purified with absolute ethanol for several times. The obtained claybank liquid was dried at 60 °C for 48 h in a vacuum oven, which was the ionic liquid precursor (step 2). Finally, the pyrolysis process was employed to anchor hm-C(CN)₃ nanosheets onto the surface of the nanopores of template via the CVD method at 650 °C under high-purity nitrogen environment (step 3).

This schematic illustration of preparation route (Figure R1) has been added in lines s111-s134 in the supporting information (renamed as Figure S3, page S12, lines s209-s211).

The pyrolytic temperature for bulk hm-C(CN)₃ sample under N₂ carrier gas was determined via TGA (Figure R2). The TGA profile discloses that heating the ionic liquids at 300 °C results in the formation of partially solidified liquid-solid intermediate and heating in the range of 450 to 550 °C gives solid products. Polymerization of [C(CN)₃]⁻ in the temperature range of 300 to 400 °C may follow a similar dynamic cyclotrimerization reaction to the condensation of aromatic nitriles, cyanamide, and acetylenes, which is accompanied by the decomposition of the corresponding ionic liquids cations. We prepared the bulk hm-C(CN)₃ sample with the temperature maintained at 500 °C. In preparation of GM@hm-C(CN)₃ samples, the AAM template was put vertically downstream of the oily precursor, and the annealing temperature was increased to 650 °C to optimize the formation of hm-C(CN)₃. After

considering a series of distance (5, 10, 15, 20 cm) from the reaction precursor, we found that the 15 cm was the optimal separation. Energy dispersive X-ray (EDX) analysis result reveals that hm-C(CN)₃ bulk sample and nanosheets derived from the ionic liquid precursor share the same C/N ratios (~ 58 at% for C and ~ 42 at% for N).

Figure R2. Scanning thermal gravimetric analysis (TGA) of the ionic liquid precursor for hm-C(CN)₃ sample.

In supporting information, the TGA curve of the precursor (Figure R2) is inserted and renamed as Figure S2 and the corresponding discussion has been added into page S7, lines s134-s147. Simultaneously, some necessary descriptions are inserted into page 4 and 5, lines 78-91.

(2) Using nanostructures to increase light absorption is not new, as this has been

employed in solar cell technologies and studied fundamentally previously. If the authors believe that this is the first time that this approach has been used in photocatalysis, this point needs to be emphasized in the paper and supported with sufficient background.

Answer: We completely agree with the reviewer's comment that using nanostructures to increase light absorption have been applied in solar cells and studied fundamentally previously. However, in our ingenious design, the special micro grid mode resonance structure makes it facile to obtain hm-C(CN)₃ nanosheets with smaller size via in-situ pyrolysis of ionic liquid besides improving solar energy absorption. The sharp decrease in size of hm-C(CN)₃ material brings about more active sites, which is very beneficial for photocatalytic hydrogen evolution. In our work, the special micro grid mode resonance structure is not only a light absorption enhancer but also a material template for size controlling. This is quite different from those applications in solar cells. To the best of our knowledge, it is the first time to apply this special micro grid mode resonance structure incorporated with hm-C(CN)₃ nanosheets for efficient photocatalytic hydrogen evolution. The sufficient background introduction and relevant references have been added into the manuscript [such as, 1). Lee, J. Y. *et al*, Nature, 460, 498-501, 2009, doi:10.1038/nature08173; 2). Tiwari, J. N. *et al*, Nature communications, 4, 2221, 2013, doi: 10.1038/ncomms3221; 3). Brongersma, M. L. *et al*, Nature materials, 13, 451-460, doi:10.1038/nmat3921, 2014; 4). Guo, C. F. *et al*, Light: Science & Applications, 3, e161, 2014, doi:10.1038/lssa.2014.42].

The necessary background descriptions are added in page 3 lines 61-69. The

previously related reports have been added in reference section (refs. 15-18).

(3) The mechanistic insight could be impactful, but the assertion that a singlet-triplet conversion plays a major role in hydrogen production needs further support. Additionally, a comment on if/how this finding could be extended beyond their specific material system would be useful.

Answer: In order to support our assertion about singlet-triplet conversion, the streak image and the time-dependent intensities of the prompt and delayed fluorescence components are shown in Figure R3. The photoluminescence spectrum is resolved into prompt and delayed components. Figure R3 shows a streak image of hm-C(CN)₃ nanosheets, which provides a visual image of time-dependent intensities of prompt and delayed fluorescence components. The intense emissions ($t < 50$ ns) correspond to the prompt component, and the long tail emissions correspond to the delayed component. The prompt component comes from the singlet fluorescence, and the delayed component can be assigned to triplet fluorescence via the intersystem crossing (ISC) from singlet to triplet excited states. A slight redshift between the prompt and delayed components is observed, which can be explained by the fact that the delayed fluorescence is generated immediately after the ISC process. Therefore, singlet-triplet conversion can be confirmed by this typical difference in transient photoluminescence characteristics.

Figure R3. Streak image and photoluminescence spectra of hm-C(CN)₃ nanosheets at 298 K.

These results and corresponding discussion are added into page 14 and 15, lines 301-316. The Figure R3 is added as Figure 6d.

The singlet-triplet conversion is intuitively demonstrated via an energy diagram in Figure R4. The activation energy of the ISC (ΔE_{ST}) is about 0.07 eV, which is proportional to the exchange energy between the singlet and triplet energy level. The reverse ISC rate constant (k_{RISC}) can be estimated from the experimentally observable rate constants and the photoluminescence quantum efficiencies of the prompt and delayed components using the following equation $k_{RISC} = \frac{k_p k_d \phi_d}{k_{ISC} \phi_p}$, where $k_p = 3.16 \times 10^6/s$ and $k_d = 7.46 \times 10^3/s$ are the constant of the prompt and delayed fluorescence components, respectively. $k_{ISC} = 1.96 \times 10^6/s$ is the ISC rate constant from singlet to triplet states, and $\phi_p = 3.6\%$ and $\phi_d = 5.9\%$ are the photoluminescence quantum efficiencies of the prompt and delayed components. The reverse ISC rate constant $k_{RISC} = 1.97 \times 10^4/s$, is smaller than $k_{ISC} = 1.96 \times 10^6/s$, which indicates that larger number of excited carriers are transferred from singlet to triplet excited states.

Figure R4. Energy diagram for singlet-triplet conversion.

These results and corresponding discussion are added into page S30 lines s537-s550, as Figure S21.

In addition, supplementary description, i.e., "Through the introduction of singlet and triplet excited states, the carrier recombination lifetime is efficiently extended from ns to μ s via singlet-triplet conversion, which can prompt more electrons transferring to active sites to participate in HER. This novel strategy allows for manipulation of carrier recombination and can be generalized to other photocatalytic materials to enhance HER performance." has been added into page 15 lines 316-321.

Comment 4. *How would you characterize the overall quality and completeness of the work?*

Reply: The question can be considered from five parts.

(1) *The quality of writing needs significant work, which makes determining the quality of the rest of the work harder to assess. Overall, it seems to be a very interesting*

material with several good experiments to highlight this (hydrogen evolution, ESR, DFT calculations, and absorption measurements), but several pieces feel unfinished. There needs to be a figure with the material, not just the repeating unit of 3 triazine rings. It is unclear to the readers exactly how those are connected.

Answer: After considering the reviewer's suggestion, some supporting results are added to clarify fuzzy assertion (marked by red). In addition, some unclear descriptions are revised. The schematic figure for structure of this material has been added and shown in Figure R5.

Figure R5. Schematic structure of hm-C(CN)₃ material.

The revised figure has been added into manuscript and renamed as Figure 1c. Some necessary descriptions are added into page 6 lines 112-115 and figure caption.

(2) Additionally, either the synthesis must be cited if it has been performed before, or if a new synthetic procedure, a proposed synthetic route should be provided in the SI.

Answer: In this manuscript, referring to previous report [Dai, S. *et al*, *Adv. Mater.* 22, 1004 (2010)], some crucial procedures and experimental conditions are modified to obtain high-quality hm-C(CN)₃ sample. The preparation route and the corresponding descriptions are provided in Figure R1, see reply to question 3 part (1).

Previous report has been cited as ref 19 and synthetic route (Figure R1) has been added into page S6 and S7, lines s111-s134 in the supporting information (renamed as Figure S3, lines s210-s211).

(3) It also must be clarified if the central carbon is a radical. If so, that must be drawn, if not, then a hydrogen must be shown in the Figure 4a and Figure 5a and b.

Answer: For pristine hm-C(CN)₃ material, the central carbon is a radical, which is marked by "•" as shown in Figure R5, R6, R7.

Figure R6. The charge density distribution of hm-C(CN)₃ nanosheets.

Figure R7. Schematic configuration-coordinate diagrams for HER mechanism.

For the photocatalytic hydrogen evolution as shown in Figure 6a and 6b, it is worth noting that the radical carbon sites will be changed with H* adsorption and hydrogen production. To avoid misunderstanding, the radical carbon has been marked by "." as shown in Figure R7.

The revised Figures R5, R6, R7 are inserted into manuscript and renamed as Figure 1(c), Figure 5(b) and Figure 6, respectively.

(4) The manuscript seems lacking in references to prior work, especially with regard to half-metallic materials and previous uses of microstructures for light absorption.

Answer: Some previous studies on half-metallic materials have been cited, Such as 1) Lee, E. C. et al. Chem.-Eur. J. 16, 12141-12146, doi:10.1002/chem.201000858 (2010). 2) Rajca, A. Chem. Rev. 94, 871-893, DOI: 10.1021/cr00028a002 (1994). 3) Liu, L. Z. et al. Appl. Phys. Lett. 106, 132406, doi: 10.1063/1.4916814 (2015).

The previous studies on using of microstructures for light absorption have been cited, see reply to question 3 part (2).

These studies have been cited and inserted in reference section (refs 11-13 and refs15-18).

(5) Additionally, the manuscript is lacking in experimental support for the proposed triplet-singlet conversion. Overall, the manuscript lacks a clear message as the importance of half-metallicity in separating photogenerated excitons is highlighted at the beginning, and the importance of triplet excitons is emphasized at the end. The reader is left confused as to if and how these two ideas are connected, if at all.

Answer: The transient photoluminescence is carried out to support our proposed triplet-singlet conversion, please see reply to question 3 Part (3) and Figure R3 and R4. It is well known that increasing carrier separation and transfer plays a critical role in improving HER performance. In our prepared half-metallic C(CN)₃ materials, the carrier recombination lifetime is extended from ns to μs via singlet-triplet conversion, which will greatly enhance the probability of electrons transferring to active sites to

participate in HER.

The corresponding discussions are added into page 15 lines 316-321.

Comment 5. *Are the conclusions adequately supported by the data?*

Reply: The question can be considered from seven parts.

(1) To properly characterize the materials presented, more data would be helpful. The material is synthesized through solution chemistry, dissolved in ethanol and concentrated, and then heated at 650°C to adhere to the microstructured template. Since carbon nitride is made from a variety of carbon- and nitrogen-rich materials at this temperature, it seems plausible that this last heating step could be altering the structure to the graphitic carbon nitride form. However, authors only present FTIR and ¹³CNMR of the solution processed material prior to heating. This is also when a full picture of the material would be helpful, because the structure drawn on the ¹³C NMR should have 3 carbon peaks, not 2.

Answer: Thank the reviewer for this significant suggestion. We reorganized the article according to the reviewer's suggestion and provided more proof experiments. The ¹³C NMR results of bulk (without template) and the small size carbon nitride nanosheets (removed template) are shown in Figure R8 for comparison. The very weak peak 3 of bulk hm-C(CN)₃ gets much stronger in nanosheets due to their broken symmetry induced by AAM template.

Figure R8. ^{13}C NMR spectra of hm-C(CN)_3 bulk sample and R@hm-C(CN)_3 nanosheets.

The revised part was added in page S17, lines s307-s310, renamed as Figure S8.

(2) Although TEM and diffraction data is presented for both pre- and post-heated material. It would be interesting to form carbon nitride in the micro grid structure and compare the hydrogen efficiencies to confirm there is a difference. It appears that the half-metallic character of the material is supported by ESR measurements, the magnetic hysteresis shown in the SI, and DFT calculations. But it would be useful for the reader if given some landmarks for typical hysteresis values.

Answer: MG@hm-C(CN)_3 sample shows a much higher hydrogen production rate of

1009 $\mu\text{mol g}^{-1} \text{h}^{-1}$, nearly 62.8 times higher than that of pristine $\text{g-C}_3\text{N}_4$ sheets. After removal of the nanotubes [named as R@hm-C(CN)_3], the photocatalytic activity drops obviously, but still much higher than that of bulk hm-C(CN)_3 sheets. This is because the hm-C(CN)_3 nanosheets of much smaller size (20 ~ 60 nm, Figure S19) have more active sites than bulk hm-C(CN)_3 sample although the capture of light reduces without micro grid structure.

The corresponding discussion is located in page 11 lines 229-236.

To obtain insights into the magnetic origin of hm-C(CN)_3 nanosheets, magnetization as function of magnetic field (M-H) is studied on superconducting quantum interference device (SQUID) at different post-processing method. As shown in Figure R9, the nonlinear hysteresis loop curve suggests that hm-C(CN)_3 nanosheets are ferromagnetic at room temperature with nonzero residual magnetization and coercivity (marked by pink line), while the $\text{g-C}_3\text{N}_4$ displays obvious diamagnetic characteristics. After hm-C(CN)_3 nanosheets soaked in dilute hydrochloric acid solutions for 2 hours, the saturation magnetization decreases from 0.0038 to 0.002 emu/g but the coercive field ($H_c = 152$ Oe) remains unchanged because the exposed radical C sites are deactivated by adsorptive H^+ , indicating that the ferromagnetism in hm-C(CN)_3 nanosheets is strongly related to the radical C sites. This magnetic behavior is different from the reported ferromagnetism in $\text{g-C}_3\text{N}_4$, which is attributed to introduced defects (0.008 emu/g) or external ions (0.004 emu/g) [Chem. Sci. **6**, 273-287 (2015); Nanoscale **6**, 2577 (2014)].

Figure R9. Room temperature magnetic hysteresis loops of hm-C(CN)₃ nanosheets with different post-processing methods.

The revised part was added in page S23, lines s380-s394, and renamed as Figure S14.

(3) *The improvement in hydrogen evolution data seems well-supported. Although the claim that it has “excellent photochemical stability” after only 12 hours of monitoring doesn’t seem completely justifiable.*

Answer: In order to examine this material's photochemical stability, the eleven successive cycles of photocatalytic hydrogen evolution over MG@hm-C(CN)₃ in 44 hours was recorded and no obvious performance decay was observed, as shown in Figure R10.

Figure R10. Recycling test of photocatalytic hydrogen production over MG@hm-C(CN)₃.

The new experimental results have been added in Figure 5d. The corresponding discussion is revised in page 12 lines 243-245.

(4)The mechanistic study presented is intriguing, it seems incomplete. The hydrogen adsorption calculation is definitely interesting and helpful to identify the likely catalytic sites. But it is somewhat questionable if the unit analyzed is actually large enough to be representative of the material. It would be helpful if this was further discussed, specifically if the terminations on C4 and C6 have any effect, since in this case they seem to be radicals.

Answer: In our calculation, the supercells with different size, such as (1×1), (2×2), (3×3), (4×4) and (5×5), are all considered to stimulate this HER mechanism. Although the values of calculated adsorption energy have some differences, the physical processes and conclusions of HER are coincident. In actual HER process, the H⁺ can unavoidably adsorbed onto C4 and C6 sites, but this adsorption cannot lead to

hydrogen generation. This is because if the H^* adsorbed onto the most active termination C2 site intends to generate molecular hydrogen more efficiently, they have to bond to neighboring H^* at C1 site. So the influence of C4 and C6 terminations on HER can be neglected.

These necessary descriptions have been inserted into page 13 lines 267-271.

(5) Experimentally, the time-resolved data is incomplete for confirming the triplet-singlet conversion as a catalytically meaningful pathway. The PL lifetime presented is certainly suggestive of a triplet, but without providing a PL quantum yield, it is hard to assess what portion of excitations results in triplet formation. Further evidence would be needed to show that it is an energy crossing of the singlet and triplet levels as the reaction proceeds and not some other process such as thermally activated delayed fluorescence. If a binding event is indeed necessary to break symmetry, then performing experiments in aprotic solvents would be useful to confirm. In general, obtaining a triplet energy experimentally would be helpful in drawing an energy landscape diagram to highlight the possible photophysical processes at play.

Answer: In order to confirm the triplet-singlet conversion, the streak image and transient photoluminescence spectra are discussed in detail, see reply to question 3 part (3). In addition, the physical parameters about triplet-singlet conversion for hm-C(CN)₃ in different solvents are shown in Table R1. Here, we choose HCl·H₂O,

H₂O and ethanol as the solvent respectively to explore contribution of H⁺ adsorption to triplet-singlet conversion, because they show different proton-donating abilities. Among them, HCL is the strongest proton-donor and the proton in ethanol's hydroxyl group is far less labile than in HCL or H₂O. The triplet excited state quantum yield ϕ_d in HCl·H₂O solvent decreases obviously, because that the radical sites are covered by H⁺, leading to its spin polarization degeneration. In ethanol, the structural symmetry is affected slightly due to sharp drop in H⁺ amount, which leads to larger value of ϕ_d . These compared results disclose that triplet-singlet conversion strongly depend on H⁺ introduction in HER reaction.

Table R1. Physical parameters about triplet-singlet conversion for hm-C(CN)₃ nanosheets in different solvents.

Different solvent	τ_p /ratio[ns]	τ_s /ratio[μ s]	ϕ_p [%]	ϕ_s [%]	K_p [10^6 s ⁻¹]	K_d [10^3 s ⁻¹]	K_{ISC} [10^6 s ⁻¹]	K_{RISC} [10^4 s ⁻¹]
H ₂ O (1)	11.4	7.9	3.6	5.9	3.16	7.46	1.96	1.97
HCl·H ₂ O (2)	15.8	4.7	4.7	2.1	2.97	4.47	0.92	1.44
Ethanol (3)	10.9	8.1	3.5	6.4	3.21	7.90	2.08	2.23

These results have been added into page S34, lines s599-s608, renamed as Table S1.

In addition, the photoluminescence (PL) spectra of hm-C(CN)₃ samples in different solvents excited by the 375 nm line of a laser are acquired and shown in Figure R11. The PL peak position is blue-shifted from 546 nm (H₂O) to 524 nm (HCl·H₂O), which can be attributed to modification of radical C site by adsorbed H⁺.

This assumption also can be further confirmed by red-shifts of PL peak in ethanol, because the decreased amount of H^+ adsorbed on radical C sites leads to higher singlet-triplet conversion. These results indicate that the H^+ involved in HER plays an important role in triplet-singlet conversion, which is in good agreement with the conclusion of Table R1.

Figure R11. Photoluminescence spectra of hm-C(CN)₃ samples in different solvents.

These results have been added into page S31, lines s558-s565, renamed as Figure S22.

In order to exclude the influence of temperature on triplet-singlet conversion, the photoluminescence decay curves of hm-C(CN)₃ at 293K, 298K, 303K and 313K (according to our experimental condition) are shown in Figure R12. The coincident

photoluminescence decay curves confirm that triplet-singlet conversion cannot be affected by temperature variation in the range of 293 to 313 K.

Figure R12. Photoluminescence decay curves of hm-C(CN)₃ at different temperatures.

These results have been added into page S32, lines s569-s573, renamed as Figure S23.

(6) The authors need to explain the origin of their assignment of the positive feature in the time-resolved absorption measurement as a trapped electron. Where does this assignment come from? The long-lived species could also be triplet absorption. The importance of this data is not clearly addressed and is not tied directly to hydrogen evolution. Overall, a remaining question is the relative importance of triplet excitons

versus separated charge carriers in the production of hydrogen, as it was previously stated in the manuscript that the half-metallic character led to efficient separation charge carriers.

Answer: Time-resolved spectroscopic measurement has obtained great attention because it is a powerful technique to analyze the carrier dynamics mechanisms. In Figure R13, we present the time-resolved absorption spectra of hm-C(CN)₃ nanosheets at select time points from 0 μs to 9.5 μs. These spectra exhibit broad positive induced absorption features between 400 nm to 650 nm, which is attributed to intersystem crossing from triplet to singlet excited states. This result is quite in agreement with the conclusion of transient photoluminescence in Figure 6d. The observed long lifetime decay kinetics indicates that the fast separation and transfer of photon-generated carriers strongly depends on the spin energy splitting induced by its intrinsic half-metallic feature. Under the support of high conductivity, the photon-generated carriers can easily inject into the catalytic sites to combine with the metastable H* atom, leading to high-efficiency H₂ generation.

Figure R13. Time-resolved adsorption measurement for hm-C(CN)₃.

These results have been added into page S33, lines s581-s590, renamed as Figure S24.

(7) A small piece but something confusing is the calculations of crystalline versus amorphous, which is mostly in the SI. The chemical structure used for amorphous is no longer the material, it has broken bonds at random, so it is unclear what the significance is of those calculations. At very least it warrants more discussion.

Answer: The amorphous calculations are displayed for comparison with crystalline structure. This is because that some amorphous structures are observed in our experimental characterization. In order to give reader a visual comparison between crystalline and amorphous (especially for experimental researchers), this parts are

added to keep completeness of details.

Comment 6. *Comments on originality and significance.*

This material appears to be novel, however, it is unclear how original or significant the incorporation into the micro grid structure is for photocatalysis. While the mechanistic study has the potential to be significant, it is incomplete and still unclear.

Answer: Nanostructures applied to increase light absorption in solar cells have previously been studied fundamentally. However, there exist obvious differences in photocatalytic application because that numbers of exposed active sites and solar light utilization must be considered simultaneously. In our ingenious design there are two major advantages: (1) We can obtain hm-C(CN)₃ nanosheets with smaller size facilely via a simple method of in-situ pyrolysis of ionic liquid (increasing number of active sites). In addition, this microstructure designs is benefit for improving solar energy adsorption (increasing solar energy exploitation). (2) For practical applications, our designed artificial microstructure incorporated by hm-C(CN)₃ nanosheets can be easily collected without need of an additional recycle technology, which are quite different from solar cell technologies. To best of our knowledge, it is the first time to apply this special micro grid mode resonance structure incorporated with hm-C(CN)₃ nanosheets for efficient photocatalytic hydrogen evolution. The sufficient background introduction and relevant references have been added into this manuscript [such as, 1). Lee, J. Y. *et al*, Nature, 460, 498-501, 2009, doi:10.1038/nature08173; 2). Tiwari, J. N. *et al*, Nature communications, 4, 2221, 2013, doi: 10.1038/ncomms3221; 3).

Brongersma, M. L. *et al*, Nature materials, 13, 451-460, doi:10.1038/nmat3921, 2014;
4). Guo, C. F. *et al*, Light: Science & Applications, 3, e161, 2014,
doi:10.1038/lsa.2014.42].

Mechanistic study on this HER reaction are further confirmed by transient photoluminescence spectra, see above replies to questions 3-5.

The corresponding descriptions about the background are added in page 3 and 4 lines 61-69. Those previously related reports have been added in reference section (refs 15-18).

Comment 7. *Comments on the quality, clarity, and conciseness of the writing.*

The overall quality of the writing is poor. There are several typographical errors and many confusing sections. The manuscript could definitely benefit from further organization and focus on a couple key takeaway messages.

Answer: Thanks for your kind reminding. We have improved the quality of the writing and revised several typographical errors. We reorganized the article according to the reviewer's significant suggestion.

The revised parts are highlighted by red.

Comment 8. *Are the figures, tables, etc. superior quality and appropriateness? What needs to be changed?*

Reply: These questions can be considered from four parts.

(1) The figure captions are not sufficiently informative. Even if markings are explained in the text, they should also be in the figure captions to help the reader.

Answer: We have improved figure captions and the revised parts were highlighted by red. See figure caption section.

(2) Figures 1 and 3 have too many components in them (9 and 11 parts respectively). Especially when formatted into an article, these will likely be too small, the chemical structures shown in 1f are already small and blurry. Several of these images are more appropriate for the SI. Similarly, Figure 2 b and c seems redundant.

Answer: The new Figures 1, 2, 3, 4 have been changed as Figure R13, R14, R15, R16 and original superfluous figures have been added into SI.

Figure R14. Characterization of hm-C(CN)₃.

Figure R15. Electronic structure of hm-C(CN)₃ nanosheets.

Figure R16. Micro grid mode resonance based artificial microstructure.

Figure R17. Simulation of micro grid mode resonance enhanced absorption based polarization optics

(3) The caption for Figure 4a refers to the “crystal structure” of the material, but this seems to just be an optimized geometry for stacked pieces of the material.

Answer: Because $\text{hm-C}(\text{CN})_3$ is layered two-dimensional material, multilayered structure is more close to that of bulk material. In order to avoid misunderstanding, the “crystal structure” is revised as "multilayered structure".

The revised part was added in the figure caption of Figure 5b, line 491.

(4) The pieces of Figure 4a seem redundant, it would be simpler to just include one (probably the lower one is more clear). Figure 4b is a bit overwhelming and hard to work through without a detailed figure caption. The arrows on 4b are very blurry.

Answer: The Figure 4a (5a) is the adsorption potential energy profile, which reflects the optimal adsorption sites of H^+ via color difference. The corresponding free energy for hydrogen adsorption is calculated and shown in Figure 6b, which is a different physical definition from Figure 6a. The simple schematic diagram in the lower inset is removed. In addition, the arrows on 5c are revised, as shown in Figure R18.

Figure R18. Schematic configuration-coordinate diagrams for HER mechanism.

The revised Figure R18 was added into manuscript as Figure 6.

Comment 9. *Is satisfactory nomenclature used?*

I'm unsure whether "Mirco Grid Mode Resonance" should be capitalized. The nomenclature to refer to the templated material as "N-C(CN)₃-" seems unwise since it is unclear why that letter is chosen and it implies nitrogen to many readers. It is also strange that the "hm" for half metallic was dropped, since presumably, it is the same material.

Answer: Thanks for your kind suggestion. The "mirco grid mode resonance" doesn't have to be capitalized and we revised the templated material as MG@hm-C(CN)₃. We also added "hm" to all C(CN)₃.

The revised parts are highlighted by red.

Comment 10. *Are potential hazards adequately described?*

No hazards stated. Safety concerns for potassium tricyanomethanide should be stated as it has significant health and safety concerns (MSDS says fatal if inhaled or contacted with skin).

Answer: Thanks for your kind suggestion.

We added the hazards statement (Toxic by inhalation, in contact with skin and if

swallowed) for the potassium tricyanomethanide in the method section, see page S6, line s114, in supporting information.

Comment 11. *Are appropriate references provided? Are they correct?*

As mentioned above, the manuscript is lacking references for nanostructures used for increased light absorption. It would also be nice to have more background for half-metallic materials addressing other half-metallic materials that have been made and how they are characterized. A paper predicting half-metallicity for the radical polymer predicts half-metal character, but is not cited in this manuscript [Lee, E. C. et al. Chemistry – A European Journal 16, 12141-12146, doi:10.1002/chem.201000858 (2010)]. References to other systems where a crossing of the triplet and singlet energy occurs would also be helpful to support this proposed mechanism, for example, some of the early work of Turro on biradical systems might be helpful here.

Answer: Thanks for your recommended literatures and comments. We have added sufficient background and related reference into this manuscript [1). Lee, J. Y. et al, Nature, 460, 498-501, 2009, doi:10.1038/nature08173. 2). Tiwari, J. N. et al, Nature communications, 4, 2221, 2013, doi: 10.1038/ncomms3221. 3). Brongersma, M. L. et al, Nature materials, 13, 451-460, doi:10.1038/nmat3921, 2014. 4). Guo, C. F. et al, Light: Science & Applications, 3, e161, 2014, doi:10.1038/lssa.2014.42].

In addition, the literatures predicting half-metallicity for the radical polymer were also added into this manuscript [1) Lee, E. C. et al. Chemistry-A European Journal 16,

12141-12146, doi:10.1002/chem.201000858 (2010). 2) Rajca, A. Chem. Rev. 94, 871-893, DOI: 10.1021/cr00028a002 (1994). 3) Liu, L. Z. et al. Appl. Phys. Lett. 106, 132406, doi: 10.1063/1.4916814 (2015)].

Some reports of Turro on biradical systems were also cited into this manuscript.

[1). Zimmt, M. B. et al, J. Am. Chem. Soc. 106, 3363-3365, DOI:10.1021/ja00323a056 (1984). 2). Turro, N. J. et al, J. Am. Chem. Soc., 97, 3859-3862, DOI: 10.1021/ja00846a075 (1975).]

The necessary descriptions about the background are added in page 3 and 4, lines 61-69. (Those previously related reports have been added in reference section (Refs. 15-18, Refs. 11-13, Refs 30, 31)

Comment (12): Suggested Improvements

Improve the clarity of the writing and proof-read more carefully.

See the section on supporting conclusions with data.

Add safety concerns.

Refine figures and provide detailed figure captions.

Answer: Thanks for your kind suggestion. We improved the writing carefully and provided more proof data. We also added the safety concerns. Last but not least, we refined figures and provided detailed figure captions according to the suggestions of the reviewer (see above).

Response to the report of the reviewer 2

A new metal-free carbon nitride with half-metallic characteristic [hm-C(CN)₃] were incorporated into artificial nanotube array through in situ pyrolysis of imidazolium-based ionic liquid, which showed highly photocatalytic activity for hydrogen evolution. But there some questions need you answer.

Comment 1. *In Figure S13, the author reported that the N-C(CN)₃ sample had a highest solar-to-hydrogen value of 2.24%. Please provide the video with the hydrogen evolution process as a solid evidence and also revise the picture with clear mark.*

Answer: Thanks for your significant suggestion. We have uploaded the video about the hydrogen evolution process as an attachment.

The picture has been revised with clear mark (such as Figure S20 and other Figures).

Comment 2. *In Figure 4b, the author mentioned about the sample “R-C(CN)₃”. However, there is no description of this sample in the manuscript or supporting information.*

Answer: Thanks for your precise suggestion. We have inserted the description of the sample “R-C(CN)₃” in the manuscript, such as “removed AAM template of the hm-C(CN)₃ in page 11 line 232.

Comment 3. In the reviewer's opinion, the bandgap of $hm-C(CN)_3$ measured from UV-DRS spectra in Fig. 3e and Fig. S7 cannot reflect the real physical information of this composite semiconductors because of the flat absorption behavior. For example, in a more simplified mode, the reviewer simulated a UV-DRS spectra with a constant absorption coefficient of 1.2 from 200 to 1000 nm as shown in Fig. 1a. Through K-M conversion, Fig. 1b shows a flat bent curve similar to the result in the manuscript. However, the calculated bandgap can be varied significantly depending on the starting point of the tangent and its slope. In fact, all black materials with a flat absorption behavior in visible light region will exhibit similar trend. The authors should check the correctness of the results and give some explanations.

Answer: Thanks for your kind review. As the reviewer said, the black materials are not suitable to calculate band gap using UV-DRS spectra. In order to ensure correctness of our measurement, cathodic and anodic scans method was adapted to re-evaluate original bandgap value, as shown in Figure R2-R1. We find that the difference in two measured method has little influence on our original conclusion about bandgap (2.14 eV for UV-DRS spectrum and 2.19 eV for linear potential scans method).

Figure R2-R1. (a) Cathodic and (b) anodic scans for determining the CB and VB energy levels of $hm-C(CN)_3$ at 5 mV s^{-1} (ref. S4) [Yeh, T. F. *et al.* Adv. Mater. 2014,

26, 3297–3303]. Applying potentials above the CB to form an accumulation layer, or below the VB to form inversion layers, can lead to abrupt emergence of cathodic and anodic currents, respectively. The bandgap of hm-C(CN)₃ was determined to be 2.19 eV, in agreement with the value obtained from the state-of-the-art hybrid functional (HSE06) calculations.

The Figure R2-R1 and corresponding description have been added into supporting information in page S21, lines s353-s358, renamed as Figure S12 to replace original UV-DRS spectrum.

Response to the report of the Reviewer 3

The work by Zhou et al present combined theoretical and experimental studies of carbon nitride nanosheets for photocatalytic hydrogen evolution. The topic is quite interesting, but graphitic carbon nitride has been well studied for solar driven water splitting.

Answer: We agree with the reviewer's opinion completely that graphitic carbon nitride has been well studied for solar driven water splitting, especially for C_3N_4 material. It is worth noting that our newly synthetic hm- $C(CN)_3$ is completely different from well-studied C_3N_4 material in both chemical structure [see Figure R3-R1] and basic physical features. This new material used for efficient HER is reported for the first time, which shows a big photocatalytic improvement in comparison with well-studied C_3N_4 material.

Figure R3-R1. Schematic structure of (a) C_3N_4 and (b) $hm-C(CN)_3$ material.

Generally photocatalytic water splitting should semiconducting with a band gap over 1.23 eV and has favorable band position for oxygen and hydrogen evolution reaction. The current work reported a metallic carbon nitride. So I feel this could be a serious problem. The metallic C_3N_4 may be only good for electrocatalysis and the claim for photocatalysis and photovoltaics is questionable. In addition, more evidence on the C_4N_3 like structure and experimental characterization of hydrogen production are also needed. Based on these, I have to recommend the rejection for publishing in high impact Journal Nature Communications.

Answer: The half-metallic feature means that spontaneous spin polarization make electronic structure around Fermi level split into spin-up and spin-down bands. The spin-down bands above Fermi level are filled completely by spin electrons, displaying metallic feature; but the spin-down bands lower than Fermi level cannot be occupied, showing semiconducting feature, as shown in Figure 2c and 2d [Phys. Rev. Lett. 108, 197207 (2012)]. In metallic materials, the spin polarization disappears and bands of spin-down and spin up begin to coincide with each other, as shown the band structure following Figure R3-R2.

Figure R3-R2, Schematic band structure and total density of state (DOS) for metallic feature: the bands and DOS both across with Fermi level.

In fact, when its spin polarization is depressed artificially, this material will displays obvious semiconducting feature in optical excitation (that is singlet excited state), as shown in Figure R3-R3. When spin polarization is considered in optical excitation, this half-metallic feature will prompt triplet excited state formation and the bandgap value becomes smaller, as shown in Figure R3-R4. The singlet-triplet conversion will lead to improvement of electron-hole separation [see energy diagram for singlet-triplet conversion in following Figure R3-R4], which is conducive to solar exploitation in HER. This singlet-triplet conversion can be confirmed by the following transient photoluminescence characteristics in Figure R3-R5 [please see detailed description in reply to reviewer 1 question 3 part (3)]. In addition, it is important to note that if this material is metallic, no obvious photoluminescence spectra are observed, as shown in Figure R3- R5 (the description, see page 14).

Figure R3-R3, (a) Band structure and (b) total density of state (DOS) for singlet state.

Figure R3-R4. Energy diagram for singlet-triplet conversion.

Figure R3-R5. Streak image and photoluminescence spectra of hm-C(CN)_3 nanosheets at 298 K.

Subsequently, the data acquired from Figure 2c are redisplayed as a more visible format in Figure R3-R6 (a) to explain the band-gap of hm-C(CN)_3 (singlet band structure: Figure R3-R3 and R3-R4). The linear potential scans display that this half-metallic material has a band-gap of 2.19 eV (larger than 1.23 eV), in agreement with that obtained from the theory calculations, see Figure S12. The positions of CB and VB are all displayed in following Figure R3-R6(b), which indicates that this

special band structure satisfy the needs of water splitting. The position of the reduction level for H^+ to H_2 is indicated by the dashed blue line, and the oxidation potential of H_2O to O_2 is indicated by the purple dashed line that is just above the valence band. The blue and pink arrows represent the CB and VB position of half-metallic $C(CN)_3$ nanosheets respectively, which are acquired from the experiment.

Figure R3-R6. (a) Band structure of half-metallic $C(CN)_3$ nanosheets, (b) Band-edge positions of the layered hm- $C(CN)_3$ relative to the vacuum level.

To further confirm the photocatalysis, the photovoltaics was measured and shown in the following Figure R3-R7. We can see that an obvious photocurrent response at -0.2 V between light on and off can be reflected by $J - t$ curves. In order to directly display the hydrogen evolution process, a video is also provided as an attachment.

Figure R3-R7. J-t cycles with ON/OFF light of half-metallic C(CN)₃ nanosheets.

Reviewers' comments:

Reviewer #1 (Remarks to the Author):

The authors have made several revisions to their manuscript. However, key logical connections that the authors are proposing remain murky and several claims do not appear to be adequately supported by the data provided. Therefore, I cannot recommend this manuscript for publication in Nature Communications in its current form until a number of major issues are resolved. In particular, the role of the purported single/triplet conversion and the claimed half-metallic character of their material in tuning the catalytic activity are unclear. Perhaps more critically, their assertion that the ferromagnetic behavior of their material unequivocally demonstrates its half-metallicity is not compelling. It appears that the author's claim of half-metallicity in their material would be more compelling were they to provide measurements of the hallmark behavior of a true half-metallic material for which the spin-resolved photoemission of majority spins should exhibit a metallic Fermi cut-off, while the minority spin should exhibit a semiconducting or insulating gap. While the material in question may in fact behave in this way, I was not able to find sufficient data in the manuscript to demonstrate that this is so. In order for this work to be further considered for publication in its current form, it seems important for the authors to address this issue. I have outlined a number of additional concerns below that I believe would need to be addressed if this work were to be considered further.

Early predictions of half-metallicity in graphitic carbon nitride were based on the assumption that the structure was the triazine unit and not on the tri-s-triazine chemical structure which it is now commonly recognized to be. Therefore, it is unclear to what degree the authors' claims can be adequately supported by the calculations that they have presented. Do the authors have additional characterization data to support the existence of the triazine-based structure over the tri-s-triazine structure? In fact, it is unclear what chemical structure the authors actually believe their material comprises. The manuscript uses the term tri-striazine [presumably a typographical cut and paste error that was originally intended to be tri-s-triazine] to describe a network of covalently bonded triazine molecules. However, the tri-s-triazine (i.e. the heptazine unit) is not what is drawn in the manuscript. The NMR peak assignments in Figure S8 do not appear to be wholly consistent with the structure that the authors have drawn for their "hm-C(CN)₃" material. In particular, the authors claim that there is a peak at 101.718 ppm, above which they have placed a label "3" that is somehow obscured by the noise. I fail to see a peak at this position that is discernible above the noise level, when by just looking at the relative peak heights it seems like one should be able to resolve this peak if it were present. Is it not possible that these are different chemical structures?

Concerning the singlet-triplet conversion discussion: the authors claim that their time-resolved diffuse reflectance (TDR) data in Figure S24 is attributable to "intersystem crossing from triplet to singlet excited states." I fail to see what exactly about this TDR data indicates intersystem crossing from the triplet to the singlet. I see no spectral evolution. I see no isosbestic behavior. I see no evidence for the system evolving to a point of dynamic equilibrium. Furthermore, the TDR data plotted here is somewhat strange, why is there no bleaching of the ground state absorption? On a related note, the authors claim in the abstract that the title material makes "the entire solar energy utilizable for hydrogen evolution reaction" seems overblown at best. They appear to be referring to the absorption data in Figure 2e that they say was collected using an integrating sphere. How was the background/blank for these measurements carried out? Clearly, the organic material does not engage in linear photon absorption in the NIR infrared. I understand that some element of their purported micro grid mode resonance structure leads to modified opacity of the sample. However, the PL data suggest that none of the photons below ~450 nm will be used productively by the material to drive any photochemical conversion.

Reviewer #2 (Remarks to the Author):

Now, I think this paper should be published at present form.

Reviewer #3 (Remarks to the Author):

Certainly the authors have basically addressed some concerns raised by the reviewers. I appreciate the efforts by the authors and there is no doubt that the performance for hydrogen production is pretty good. However the mechanism is not fully clear and I can not believe half-metallic photocatalyst can be used for efficiently solar driven water splitting. Are there any previous works that reported metallic or half-metallic photocatalysts for hydrogen production? I would rather believe that the synthesized C₄N₃ is still semiconductor, possibly due to the defect or vacancy?

**"Half-Metallic Carbon Nitride Nanosheets with Micro Grid
Mode Resonance Structure for Highly Efficient Photocatalytic
Hydrogen Evolution"**

ID: NCOMMS-17-30419A

Response to the report of the Reviewer 1

Comment 1. *The authors have made several revisions to their manuscript. However, key logical connections that the authors are proposing remain murky and several claims do not appear to be adequately supported by the data provided. Therefore, I cannot recommend this manuscript for publication in Nature Communications in its current form until a number of major issues are resolved. In particular, the role of the purported single/triplet conversion and the claimed half-metallic character of their material in tuning the catalytic activity are unclear. Perhaps more critically, their assertion that the ferromagnetic behavior of their material unequivocally demonstrates its half-metallicity is not compelling. It appears that the author's claim of half-metallicity in their material would be more compelling were they to provide measurements of the hallmark behavior of a true half-metallic material for which the spin-resolved photoemission of majority spins should exhibit a metallic Fermi cut-off, while the minority spin should exhibit a semiconducting or insulating gap. While the material in question may in fact behave in this way, I was not able to find sufficient data in the manuscript to demonstrate that this is so. In order for this work to be further considered for publication in its current form, it seems important for the authors to address this issue. I have outlined a number of additional concerns below that I believe would need to be addressed if this work were to be considered further.*

Answer: Thanks for the reviewer's comments and suggestions. In order to further support our assertion about the half-metallic character, the spin-resolved photoemission spectra of $C(CN)_3$ sample were recorded. Figure R1 shows valence band spin-resolved photoemission spectra near Fermi level (E_F) at 300 K of synthesized hm- $C(CN)_3$ sample, evidencing considerable differences between the majority (down) and minority (up) spins. The spectrum for the majority spin extends up to E_F and shows the metallic Fermi cut-off, while that for the minority spin decreases rapidly at binding energy of ~ 0.6 eV and the spectral weight disappears very near to E_F , indicating the insulating gap. Previous theoretical calculations disclose that introduction of radical C1 site will inject a hole into original graphitic C_3N_4 structure [Du, A. J et al, Phys. Rev. Lett. **108**, 197207 (2012); Lee, E. C. et al, Chem.- Eur. J. **16** 12141 (2010)], which makes spin polarization occur at neighboring N atoms. The minority-spin states still show small spectral weight in higher-binding-energy region where the C p orbitals (C1 site, Figure 6a) are fully occupied by both spins. The p orbital electrons of N atoms (N3, N5 and N7 site, Figure 6a) are spin-polarized, leading to ferromagnetic behavior. The spin density of N p orbital can be obtained by subtracting the minority-spin spectrum from majority-spin spectrum. The difference spectrum presented in bottom panel of Figure R1 shows the metallic Fermi cut-off at E_F and a peak feature around binding energy of 0.75 eV, which can be interpreted as the spin splitting induced by sp^2 hybridization between C and N atoms. These results demonstrated that the synthesized $C(CN)_3$ material has a predicated half-metallic feature. This experimental conclusion is well consistent with previous theoretical reports, such as Du, A. J et al, Phys. Rev. Lett.

Figure R1. Spin-resolved photoemission spectra of hm-C(CN)_3 sample near the Fermi energy (E_F) at 300 K. The photon energy was set at $h\nu=40$ eV. The bottom panel shows the difference spectrum between the majority-spin and minority-spin spectra.

The revised part was inserted into page 8 lines 162-170 and page S23 and S24, Figure R1 are renamed as Figure S13.

Comment 2. Early predictions of half-metallicity in graphitic carbon nitride were

based on the assumption that the structure was the triazine unit and not on the tri-s-triazine chemical structure which it is now commonly recognized to be. Therefore, it is unclear to what degree the authors claim can be adequately supported by the calculations that they have presented. Do the authors have additional characterization data to support the existence of the triazine-based structure over the tri-s-triazine structure? In fact, it is unclear what chemical structure the authors actually believe their material comprises. The manuscript uses the term tri-s triazine [presumably a typographical cut and paste error that was originally intended to be tri-s-triazine] to describe a network of covalently bonded triazine molecules. However, the tri-s-triazine (i.e. the heptazine unit) is not what is drawn in the manuscript. The NMR peak assignments in Figure S8 do not appear to be wholly consistent with the structure that the authors have drawn for their “hm-C(CN)₃” material. In particular, the authors claim that there is a peak at 101.718 ppm, above which they have placed a label “3” that is somehow obscured by the noise. I fail to see a peak at this position that is discernible above the noise level, when by just looking at the relative peak heights it seems like one should be able to resolve this peak if it were present. Is it not possible that these are different chemical structures?

Answer: We completely agree with the reviewer's comment that graphitic carbon nitride of half-metallicity has the structure based on triazine unit and not on tri-s-triazine chemical structure. After careful verification, we found that "triazine" was miswritten as "tri-s-triazine" in our manuscript. In fact, all experimental characterizations confirm that our synthesized samples only have triazine unit. All the

miswritten "tri-s-triazine"s have been changed to "triazine". In addition, to avoid misunderstanding about the structure of hm-C(CN)_3 , the NMR spectra were measured again to display peak 3 more clearly. Figure R2 shows the new NMR spectra in which the weak peak 3 is clearly visible.

Figure R2. ^{13}C NMR spectra of hm-C(CN)_3 bulk sample and R@hm-C(CN)_3 nanosheets.

The corresponding revisions in the manuscript have been marked by red in line 174, and the Figure R2 was renamed as Figure S8 in page S18.

Comment 3. Concerning the singlet-triplet conversion discussion: the authors claim that their time-resolved diffuse reflectance (TDR) data in Figure S24 is attributable to “intersystem crossing from triplet to singlet excited states.” I fail to see what exactly about this TDR data indicates intersystem crossing from the triplet to the singlet. I see

no spectral evolution. I see no isosbestic behavior. I see no evidence for the system evolving to a point of dynamic equilibrium. Furthermore, the TDR data plotted here is somewhat strange, why is there no bleaching of the ground state absorption?

Answer: We deeply appreciate the reviewer's comments. In order to clarify the conversion from singlet to triplet excited states, the time-resolved absorption spectra of hm-C(CN)₃ nanosheets were carried out again. The dynamics of photoexcited carriers on different lifetime scales indicates the different physical mechanisms in hm-C(CN)₃ nanosheets. In general, transient absorption (TA) spectrum consists of multiple components including ground-state bleaching (GSB, $\Delta T/T > 0$), excited-state absorption (ESA, $\Delta T/T < 0$), and stimulated emission (SE, $\Delta T/T > 0$) from different excited states with their characteristic lifetime parameters. Figure R3a shows the TA spectra recorded at different time delays with the pulse excitation at 360 nm. A photoinduced bleaching (PIB) signal appears instantaneously with the incidence of pump pulse. Following the recovery of the initial PIB signal on ps time scale, the signal of photoinduced absorption (PIA) emerges in the spectral range > 480 nm. The PIA signal peaks at a time delay of 50 ps and recovers fully on a time scale of 2.1 ns. Surprisingly, a late-stage PIB signal becomes pronounced after the recovery of the PIA signal at time delay > 2.1 ns. Subsequently, in Figure R3b, we analyze the fs-resolved kinetic curves for the assignment of different TA components. The beginning curve is enlarged and shown in the inset, exhibiting a fast recombination dynamics on ps scale. Such a fast recombination ($\tau \approx 0.9$ ps) is commonly observed during the thermalization of hot carriers whose lifetime is mainly governed by the

electron-phonon (e-ph) interaction. In addition, temporal evolution of the PIA signal amplitude shows a slightly increase with longer probe time. The kinetic curve can be fitted by a biexponential decay function. The life time of the fast component is ~ 354 ps, comparable to the time parameter of carrier recombination [Nano. Lett. **15** 4650-4656 (2015)]. However, this fast component ($\tau \approx 354$ ps) of amplitude ratio less than 25% plays a much less significant role in the recovery of the photoexcited carriers. A primary channel is the much slower component with a long lifetime of 2.8 ns whose amplitude ratio is in excess of 75%. The slow dynamics may be related to the singlet-triplet intersystem crossing since the slow component extends to the buildup of a long-lived PIA signal. To further understand the relation between the PIA feature on the long time scale and singlet-triplet conversion, Figure R3c shows the counterplots of TA signal as a function of probe wavelength and delay time measured with ns resolution. Following the recovery, the PIB signal of broadband coverage (>550 nm) gradually builds up on a time scale from several ns to 100 μ s. It is reasonable to connect the long-lived feature to the triplet population as the recombination of triplet with ground state is spin forbidden. For example, the kinetics probed at 650 nm (acquired from gray line) displays that the recovery lifetime of the PIB signal is ~ 9.2 μ s, besides the onset of bleaching ($\tau \approx 2.5$ ns). Such a long-lived component can be regarded as a solid evidence for the triplet generation. The TA feature for the long-lived triplets in the visible range manifests as the PIB, suggesting the major ESA of triplets is probably in the infrared range. The measured onset lifetime ($\tau \approx 2.5$ ns) is close to that of the major carrier relaxation channel ($\tau \approx 2.8$ ns) uncovered in fs-resolved TA spectroscopy, implying a high efficiency of singlet-triplet

crossover. Theoretically, the rate of intersystem crossing is sensitive to energy difference between the singlet and triplet states (ΔE_{ST}). Associated with transient fluorescence feature in Figures S22-S24 and Table S1, the splitting between singlet and triplet states has been calculated to be ~ 70 meV, and the efficiency of singlet-triplet conversion can be roughly estimated to be $\sim 65\%$. Therefore, the schematic diagram of the kinetic model describing carrier dynamics is displayed in Figure R3d. The photoexcitation generates a hot carrier that cools down to the band edge in subps (~ 0.9 ps). A small part of the cooled carriers may recombine with the ground state through a fast channel with lifetime of ~ 354 ps while the rest of carriers undergo the intersystem crossing from singlet to triplet states (~ 2.8 ns). Carriers occupying the triplet states recombine with the ground state slowly due to the spin forbidden effect (~ 9.2 μ s).

Figure R3 (a) Transmission spectra ($\Delta T/T$) at different time delays. (b) The curves probed at 650 nm, and the inset is the enlarged onset curve. (c) Nanosecond-resolved TA spectra in the decay range up to 100 μ s. (d) Schematic model of triplet dynamics.

The discussion about transient absorption (TA) spectra was inserted into page S39-S41, and the Figure R3 is renamed as Figure S25.

Comment 4. Reply: This question can be considered from three parts.

(1) On a related note, the authors claim in the abstract that the title material makes “the entire solar energy utilizable for hydrogen evolution reaction” seems overblown

at best. They appear to be referring to the absorption data in Figure 2e that they say was collected using an integrating sphere. How was the background/blank for these measurements carried out?

Answer: (1) In our measurement, the background/blank effect has been considered. The diffuse reflectance absorption spectra (DRS) of the samples were recorded by a UV-vis spectrophotometer (Varian Cary 5000) equipped with an integrating sphere attachment with BaSO₄ as a reference. In the DRS measurement, there are a large number of scattering spots in the sample so that the absorption of sample cannot be directly tested. The measurement of diffusion reflection in the sample is used to calculate the absorption according to the Kubelka-Munk (KM) theory:

$$\frac{K}{S} = \frac{(1-R_{\infty})^2}{2R_{\infty}} = F(R_{\infty}) \quad (1)$$

Where the R_{∞} is the reflectance, K and S are the absorption coefficient and scattering coefficient respectively. F is the emission function of KM. The equation (1) can be rewritten as follows:

$$\lg F(R_{\infty}) = \lg K - \lg S \quad (2)$$

If S, basically, is independent of the wavenumber, the scattering effect only influences the spectrum line along the longitudinal axis. Under this condition, $F(R_{\infty})$ represents the real absorption spectrum of sample. Thus the reflectance (R_{∞}) must be measured. The R_{∞} parameters of nonabsorbent materials, for example MgO and BaSO₄, are usually taken as references. The reflectance R'_{∞} can be obtained by the following equation:

$$R'_{\infty} = \frac{R_{\infty}(\text{sample})}{R_{\infty}(\text{reference substance})} \quad (3)$$

During the measurement, an integrating sphere attachment with BaSO₄ as a reference is used to collect the diffusive light and avoid the diffusive differences caused by the light collection process. Firstly, the diffuse reflectance absorption of BaSO₄ powder is recorded as the background/blank sample and baseline, and then the samples were put into the integrating sphere for measurement. The background/blank (BaSO₄) for these measurements is automatically carried out by the UV-vis spectrophotometer.

The above description on reflectance absorption spectra measurement has been inserted into page S9-S10.

(2) Clearly, the organic material does not engage in linear photon absorption in the NIR infrared. I understand that some element of their purported micro grid mode resonance structure leads to modified opacity of the sample.

Answer: We completely agree with the reviewer's comment that the organic material does not engage in linear photon absorption in the NIR infrared. Generally speaking, both ways of reducing reflections and increasing effective optical path length of an incident light ray are utilized to increase light absorption. Here, in the fabrication process, hm-C(CN)₃ nanosheets are randomly incorporated into AAM to construct cylindrical resonators (void resonators), in which the inner walls are rough. Then, this structure can be simply regarded as light scatters off nanoscale voids within a high-index dielectric or high-index dielectric surrounded by a low-index embedding medium, typically air or water. The scattering properties of cylindrical resonators can be described with analytical Mie theory. In Mie theory the incident, internal, and

scattered fields are decomposed into a set of vector harmonics \vec{M}_m and \vec{N}_m , which are indexed by their azimuthal phase dependence $e^{im\varphi}$. At normal incidence the scattered fields are purely transverse electric (TE) or transverse magnetic (TM) with excitation coefficients a_m and b_m , respectively:

$$a_m(x) = \frac{n_r J'_m(x) J_m(n_r x) - J_m(x) J'_m(n_r x)}{n_r J_m(n_r x) H'_m(x) - J'_m(n_r x) H_m(x)} \quad (1)$$

$$b_m(x) = \frac{J'_m(x) J_m(n_r x) - n_r J_m(x) J'_m(n_r x)}{J_m(n_r x) H'_m(x) - n_r J'_m(n_r x) H_m(x)} \quad (2)$$

where J_m and H_m are respectively Bessel and Hankel functions of the first kind and primes denote derivatives with respect to the argument. The relative refractive index n_r and relative size x are given by

$$n_r = \frac{k_{\text{int}}}{k_{\text{emb}}} = \frac{n_{\text{int}}}{n_{\text{emb}}}, \quad x = k_{\text{emb}} r_0 = \frac{2\pi n_{\text{emb}} r_0}{\lambda} \quad (3)$$

Here n_{emb} and n_{int} are the refractive indexes of the embedding medium and the resonator respectively, k is the wavevector, r_0 is the cylinder radius, and λ is the free space wavelength. The scattering and extinction properties of single resonators can be easily determined once the Mie coefficients are calculated. The cylindrical resonators make incident lights travel with multiage scattering and coherent superposition, which can efficiently increase the near infrared (NIR) absorption, as shown in Figure 3e. When hm-C(CN)₃ nanosheets are incorporated in the micro grid mode resonance structure, as shown in Figure R4, the photon absorption behaves in a linear way. The FDTD simulations disclose that the distribution of electric field at nanotube surface is enhanced. Moreover, when the natural light projected at the interface of the media undergoes reflection and refraction, light vector can be decomposed into two parts (as

shown in Figure 4b and Figure S16). One part parallel to the incident surface is called the P wave (marked by blue arrows: E_{1p}), and the other part perpendicular to the incident surface is called the S wave (marked by red arrows: E_{1s}). In this special columnar micro grid structure, the incident natural lights are all expressed as P-wave on each plane perpendicular to the nanotube inner walls. Therefore, NIR absorption is enhanced via P wave behavior and cylindrical resonators, leading to a linear NIR absorption, as shown in Figure 3e. Besides, a hollow ring waveguide structure with AAM and air (water) as low refractive index medium is formed. Thus, all the light coupled into hm- C_4N_3 experiences multiple reflections in the waveguide, which increase the effective optical path length of the light.

Figure R4. Absorption zones of P-wave versus wavelength and incident angle for smooth (a) and rough (b) inner walls in nanotube arrays.

The discussion about near infrared (NIR) absorption was inserted into page S28-S30 and page 10 lines 214-219, and the Figure R4 is renamed as Figure S17.

(3) However, the PL data suggest that none of the photons below ~450 nm will be

used productively by the material to drive any photochemical conversion.

Answer: In order to clarify the contributions from lights of different wavelengths to the hydrogen evolution, photocatalytic H₂ evolution and On/Off photocurrent curves were carried out under different irradiation lights. In our measurement, 380/750 nm long-wave-pass and 380/750 nm short-wave-pass were chosen to obtain the required Vis + NIR light, pure NIR light, pure UV light and UV + Vis light, and the light intensity was calibrated to 100 mW cm⁻². Figure R5a shows the hydrogen evolution rates with/without near infrared (NIR) or ultraviolet (UV) lights, which indicates that the hydrogen evolution rates decrease obviously in the absence of NIR or UV lights. These results strongly evidence that the entire solar energy can make contributions to the hydrogen evolution. In addition, Figure R5b gives amperometric I-t curves of hm-C(CN)₃ photocathode at -0.2 V under the chopped light irradiation with and without NIR/UV light. Obviously, the photocurrent responds fast to each switch event in both cases. With the continuous light irradiation, hm-C(CN)₃ maintains reproducible photocurrent without detectable decay, indicating of its PC stability. Note that the photocurrent enhances with the addition of NIR light, whereas the bare NIR light irradiation does not yield any apparent photocurrent. Theoretically, hm-C(CN)₃, a semiconductor of band gap at ~ 550 nm, cannot be activated to generate photoelectrons under bare NIR light irradiation. It is reasonable to connect that the NIR light induces a mediating effect to further promote the photoelectron output. After hm-C(CN)₃ sample irradiated by NIR light for 30 min, its temperature increases from 25 °C to 40 °C due to the thermal effect of NIR light, with a remarkable rise in the electric conductivity (see Figure R6). The higher

electric conductivity can promote the migration of photogenerated carriers and lead to the higher carrier mobility. Thus, the NIR-enhanced photocurrent can be ascribed to the NIR-induced thermal effect, in good accordance with Figure R5b. Besides, it is worth noting that bare UV light irradiation (larger than band gap E_g) produces a small photocurrent, but a remarkable decline in photocurrent occurs when the UV light contribution is removed. Under UV light irradiation, large numbers of photons are absorbed to generate carriers. The carriers occupied at higher energy states will relax to conduction band edge via nonradiative recombination (see analysis about time-resolved absorption spectra in Figure R3), and then recombine with the ground states. In this recombination process, a small part of carriers will join in the hydrogen evolution reaction. Therefore, photocurrent induced by bare UV light irradiation can also be observed.

Figure R5 (a) Photocatalytic H₂ evolution from water under different irradiation lights on MG@hm-C(CN)₃. (b) Amperometric I-t curves of hm-C(CN)₃ photocathode at the potential of -0.2 V (vs Ag/AgCl) under the chopped on-off cycles of different light sources: the white lights with and without NIR/UV component, and pure NIR/UV light.

Figure R6 The electrical conductivity of hm-C(CN)₃ nanosheets versus temperature curve.

The necessary descriptions have been inserted into page 12 lines 244-250 and page S33-S34. The Figure R5 and Figure R6 were renamed as Figure S20 and Figure 5a, respectively.

End

Response to the report of the reviewer 3

Comment: Certainly the authors have basically addressed some concerns raised by the reviewers. I appreciate the efforts by the authors and there is no doubt that the performance for hydrogen production is pretty good. However the mechanism is not fully clear and I cannot believe half-metallic photocatalyst can be used for efficiently solar driven water splitting. Are there any previous works that reported metallic or half-metallic photocatalysts for hydrogen production? I would rather believe that the synthesized C_4N_3 is still semiconductor, possibly due to the defect or vacancy?

Answer: Thanks for the reviewer's comments. We completely agree on the reviewer's opinion that this material is a semiconductor with half-metallic feature, but not a half-metallic photocatalyst. A semiconducting material with half-metallic feature means that spontaneous spin polarization makes electronic structure around Fermi level split into spin-up and spin-down bands. Essentially, this material also belongs to semiconductor, but special spin state electron can display metallic feature. In addition, the photocatalytic property of half-metallic materials has been reported in some previous studies. For example, $La_{0.7}Sr_{0.3}MnO_3$ and Fe_3O_4 with typical half-metallic feature, confirmed by [Nature **392**, 794-796 (1998); Phys. Rev. Lett. **87** 026601 (2001)], can be used to photocatalytic hydrogen evolution or hazardous substance degradation [Nanoscale **4**, 5202 (2012); New J. Chem. **39**, 2413 (2015); Mater. Lett. **131**, 125 (2014); Ultrason. Sonochem. **20**, 1419 (2013); Sep. Purif. Technol. **134**, 12–19 (2014)]. However, both of them have the low photocatalytic activity, which

limits them to practical applications. Therefore, half-metallic material with high photocatalytic performance has always been the subject of intense research activities. In 2011, the $C(CN)_3$ material was synthesized for the first time and reported in *Adv. Mater.* **22**, 1004 (2010). Then, half-metallic C_4N_3 material with larger band gap (~ 2.2 eV) was predicated [*Phys. Rev. Lett.* **108**, 197207 (2012)], which provides the possibility for hydrogen evolution. Through the introduction of singlet and triplet excited states, the carrier recombination lifetime is efficiently extended from ns to μ s via singlet-triplet conversion, which can prompt more electrons to transfer to active sites for HER. Subsequently, we confirm half-metallic feature and its contribution to singlet-triplet conversion point by point.

Discussion about half-metallic characteristic: As a direct evidence for the half-metallic feature, the spin-resolved photoemission spectra of hm- $C(CN)_3$ sample were carried out. Figure R31 shows valence band spin-resolved photoemission spectra near Fermi level (E_F) at 300 K of synthesized hm- $C(CN)_3$ sample, evidencing considerable differences between the majority (down) and minority (up) spins. The spectrum for the majority spin extends up to E_F and shows the metallic Fermi cut-off, while that for the minority spin decreases rapidly at binding energy of ~ 0.6 eV and the spectral weight disappears very near to E_F , indicating the insulating gap. Previous theoretical calculations disclose that introduction of radical C1 site will inject a hole into original graphitic C_3N_4 structure [Du, A. J et al, *Phys. Rev. Lett.* **108**, 197207 (2012); Lee, E. C. et al, *Chem.- Eur. J.* **16** 12141 (2010)], which makes spin polarization occur at neighboring N atoms. The minority-spin states still show small

spectral weight in higher-binding-energy region where the C p orbitals (C1 site, Figure 6a) are fully occupied by both spins. The p orbital electrons of N atoms (N3, N5 and N7 site, Figure 6a) are spin-polarized, leading to ferromagnetic behavior. The spin density of N p orbital can be obtained by subtracting the minority-spin spectrum from majority-spin spectrum. The difference spectrum presented in bottom panel of Figure R1 shows the metallic Fermi cut-off at E_F and a peak feature around binding energy of 0.75 eV, which can be interpreted as the spin splitting induced by sp^2 hybridization between C and N atoms. These results demonstrated that the synthesized $C(CN)_3$ material has a predicated half-metallic feature. This experimental conclusion is well consistent with previous theoretical reports, such as Du, A. J et al, Phys. Rev. Lett. **108**, 197207 (2012); Lee, E. C. et al, Chem.- Eur. J. **16** 12141 (2010).

Figure R31. Spin-resolved photoemission spectra of hm- $C(CN)_3$ sample near the Fermi energy (E_F) at 300 K. The photon energy was set at $h\nu=40$ eV. The bottom panel shows the difference spectrum between the majority-spin and minority-spin spectra.

The revised part was inserted into page 8 lines 162-170 and page S23 and S24, Figure R31 are renamed as Figure S13.

Based on previous theoretical predications [Phys. Rev. Lett. **108**, 197207 (2012); Appl. Phys. Lett. **106**, 132406 (2015)], the half-metallic characteristic mainly depends on introduction of active site C1 (see, Figure 6a). In hydrogen evolution reaction, the active sites C1 and C2 will be covered by adsorptive H^* , leading to decrease in spin splitting (see detailed discussion about Figure 6). This theoretical prediction can be confirmed by magnetic intensity changes of samples treated with different post-processing methods. Magnetization versus magnetic field (M-H) curves for different samples were recorded on superconducting quantum interference device (SQUID) to obtain insights into the magnetic origin of $hm-C(CN)_3$ nanosheets. As shown in Figure R32, the nonlinear hysteresis loop curve suggests that $hm-C(CN)_3$ nanosheets are ferromagnetic at room temperature with nonzero residual magnetization and coercivity (marked by pink line), while the $g-C_3N_4$ displays obviously diamagnetic characteristics. With $hm-C(CN)_3$ nanosheets soaked in dilute hydrochloric acid solution for 2 hours, the saturation magnetization decreases from 0.0038 to 0.002 emu/g, but the coercive field ($H_c = 152$ Oe) remains unchanged, because the exposed radical C sites are deactivated by adsorptive H^+ . These results evidence that the half-metallic feature in $hm-C(CN)_3$ nanosheets is strongly related to the active radical C sites (C1 and C2, Figure 6). Thus, it is reasonable to connect the half-metallicity to the hydrogen evolution as the carrier recombination lifetime is efficiently extended from ns to μs via singlet-triplet conversion, which can prompt more electrons to transfer to active sites for HER.

Figure S32. Room temperature magnetic hysteresis loops of the hm-C(CN)₃ nanosheets with different post-processing methods and the g-C₃N₄ as a reference.

These descriptions and the Figure S14 can be found in page S25

Discussion about half-metallic contribution to hydrogen evolution: In order to confirm the half-metallic contribution more directly, the hm-C(CN)₃ nanosheets were annealed at 573 K in H₂ environment. Under this condition, the active sites, leading to spin polarization, decrease sharply, as shown in Figure R32. As a result, the hydrogen production rate is reduced to 23 %, compared with unannealed sample (see Figure R37). This solidly evidences that the half-metallicity induced by active sites C1 and C2 plays an important role in hydrogen production reaction (Figure 6). Interestingly, those active sites involved in hydrogen production reaction are able to determine the emergence/disappearance of half-metallic feature [Phys. Rev. Lett. **108**, 197207 (2012); Lee, E. C. et al, Chem.- Eur. J. **16** 12141 (2010); Appl. Phys. Lett. **106** 132406

(2015)]. Therefore, this novel strategy allows for manipulation of carrier recombination by singlet-triplet conversion and can be generalized to other photocatalytic materials to enhance HER performance.

Figure R33 Comparison of the photocatalytic hydrogen production rate under the white light irradiation on the samples before (Sample 1) and after (Sample 2) annealed in H₂ environment.

Discussion about half-metallic contribution to singlet-triplet conversion by transient

fluorescence: To better understand the influence of spin state and carrier recombination, the streak image and the time-dependent intensities of the prompt and the delayed fluorescence components are shown in Figure R34. The photoluminescence spectrum is resolved into the prompt and the delayed components. The streak image of hm-C(CN)₃ nanosheets provides a visual image of time-dependent intensities of the prompt and the delayed fluorescence components. The intense emissions ($t < 50$ ns) correspond to the prompt component, and the long

tail emissions correspond to the delayed component. The prompt component comes from the singlet fluorescence, and the delayed component can be assigned to triplet fluorescence originating from the intersystem crossing (ISC) from singlet to triplet excited states. A slight redshift between the prompt and the delayed components is observed, which can be explained by the fact that the delayed fluorescence is generated immediately after the ISC process. Therefore, singlet-triplet conversion can be confirmed by this typical difference in transient photoluminescence characteristics.

Figure R34. Streak image and photoluminescence spectra of hm-C(CN)_3 nanosheets at 298 K.

The singlet-triplet conversion is intuitively demonstrated via an energy diagram in Figure R35. The activation energy of the ISC (ΔE_{ST}) is about 0.07 eV, which is proportional to the exchange energy between the singlet and the triplet energy levels. The reverse ISC rate constant (k_{RISC}) can be estimated from the experimentally observable rate constants and the photoluminescence quantum efficiencies of the prompt and the delayed components using the following equation $k_{\text{RISC}} = \frac{k_p k_d \phi_d}{k_{\text{ISC}} \phi_p}$, where $k_p = 3.16 \times 10^6/\text{s}$ and $k_d = 7.46 \times 10^3/\text{s}$ are the constants of the prompt and the

delayed fluorescence components, respectively. $k_{ISC}=1.96\times 10^6/s$ is the rate constant of ISC from singlet to triplet states, and $\phi_p = 3.6\%$ and $\phi_d = 5.9\%$ are the photoluminescence quantum efficiencies of the prompt and the delayed components. The reverse ISC rate constant $k_{RISC}=1.97\times 10^4/s$, is smaller than $k_{ISC}=1.96\times 10^6/s$, indicating that larger number of excited carriers are transferred from singlet to triplet excited states.

Figure R35. Energy diagram for singlet-triplet conversion.

The physical parameters about singlet-triplet conversion for hm-C(CN)₃ in different solvents were collected and shown in Table R35. Here, we choose HCl·H₂O, H₂O and ethanol as the solvents respectively to explore contribution of H⁺ adsorption to singlet-triplet conversion because they show different proton-donating abilities. Among them, HCL is the strongest proton-donor and the proton in ethanol's hydroxyl group is far less labile than in HCL or H₂O. The triplet excited state quantum yield ϕ_d in HCl·H₂O solvent decreases obviously, because the radical sites are covered by H⁺, leading to the spin polarization degeneration. In ethanol, the structural symmetry is affected slightly due to sharp drop in H⁺ number, which results in larger value of ϕ_d .

These comparison results prove that singlet-triplet conversion strongly depends on H⁺ introduction to HER reaction.

Table R36 Physical parameters about singlet-triplet conversion for hm-C(CN)₃ nanosheets in different solvents.

Different solvent	τ_p /ratio[ns]	τ_d /ratio[μ s]	ϕ_p [%]	ϕ_d [%]	K_p [10^6 s ⁻¹]	K_d [10^3 s ⁻¹]	K_{ISC} [10^6 s ⁻¹]	K_{RISC} [10^4 s ⁻¹]
H ₂ O (1)	11.4	7.9	3.6	5.9	3.16	7.46	1.96	1.97
HCl·H ₂ O (2)	15.8	4.7	4.7	2.1	2.97	4.47	0.92	1.44
Ethanol (3)	10.9	8.1	3.5	6.4	3.21	7.90	2.08	2.23

These descriptions and the Figures S22-S24 and Table S1 can be found in page S36-S38 and S42.

Discussion about half-metallic contribution to singlet-triplet conversion by transient

absorption: In order to clarify the conversion from singlet to triplet excited states, the time-resolved absorption spectra of hm-C(CN)₃ nanosheets were carried out again. The dynamics of photoexcited carriers on different lifetime scales indicates the different physical mechanisms in hm-C(CN)₃ nanosheets. In general, transient absorption (TA) spectrum consists of multiple components including ground-state bleaching (GSB, $\Delta T/T > 0$), excited-state absorption (ESA, $\Delta T/T < 0$), and stimulated emission (SE, $\Delta T/T > 0$) from different excited states with their characteristic lifetime parameters. Figure R37a shows the TA spectra recorded at different time delays with the pulse excitation at 360 nm. A photoinduced bleaching (PIB) signal appears instantaneously with the incidence of pump pulse. Following the recovery of the

initial PIB signal on ps time scale, the signal of photoinduced absorption (PIA) emerges in the spectral range >480 nm. The PIA signal peaks at a time delay of 50 ps and recovers fully on a time scale of 2.1 ns. Surprisingly, a late-stage PIB signal becomes pronounced after the recovery of the PIA signal at time delay > 2.1 ns. Subsequently, in Figure R37b, we analyze the fs-resolved kinetic curves for the assignment of different TA components. The beginning curve is enlarged and shown in the inset, exhibiting a fast recombination dynamics on ps scale. Such a fast recombination ($\tau_1 \approx 0.9$ ps) is commonly observed during the thermalization of hot carriers whose lifetime is mainly governed by the electron-phonon (e-ph) interaction. In addition, temporal evolution of the PIA signal amplitude shows a slightly increase with longer probe time. The kinetic curve can be fitted by a biexponential decay function. The life time of the fast component is ~ 354 ps, comparable to the time parameter of carrier recombination [Nano. Lett. **15** 4650-4656 (2015)]. However, this fast component ($\tau_2 \approx 354$ ps) of amplitude ratio less than 25% plays a much less significant role in the recovery of the photoexcited carriers. A primary channel is the much slower component with a long lifetime of 2.8 ns whose amplitude ratio is in excess of 75%. The slow dynamics may be related to the singlet-triplet intersystem crossing since the slow component extends to the buildup of a long-lived PIA signal. To further understand the relation between the PIA feature on the long time scale and singlet-triplet conversion, Figure R37c shows the counterplots of TA signal as a function of probe wavelength and delay time measured with ns resolution. Following the recovery, the PIB signal of broadband coverage (>550 nm) gradually builds up on a time scale from several ns to 100 μ s. It is reasonable to connect the long-lived

feature to the triplet population as the recombination of triplet with ground state is spin forbidden. For example, the kinetics probed at 650 nm (acquired from gray line) displays that the recovery lifetime of the PIB signal is $\sim 9.2 \mu\text{s}$, besides the onset of bleaching ($\tau \approx 2.5 \text{ ns}$). Such a long-lived component can be regarded as a solid evidence for the triplet generation. The TA feature for the long-lived triplets in the visible range manifests as the PIB, suggesting the major ESA of triplets is probably in the infrared range. The measured onset lifetime ($\tau \approx 2.5 \text{ ns}$) is close to that of the major carrier relaxation channel ($\tau_3 \approx 2.8 \text{ ns}$) uncovered in fs-resolved TA spectroscopy, implying a high efficiency of singlet-triplet crossover. Theoretically, the rate of intersystem crossing is sensitive to energy difference between the singlet and triplet states (ΔE_{ST}). Associated with transient fluorescence feature in Figures S22-S24 and Table S1, the splitting between singlet and triplet states has been calculated to be $\sim 70 \text{ meV}$, and the efficiency of singlet-triplet conversion can be roughly estimated to be $\sim 65\%$. Therefore, the schematic diagram of the kinetic model describing carrier dynamics is displayed in Figure R37d. The photoexcitation generates a hot carrier that cools down to the band edge in subps ($\sim 0.9 \text{ ps}$). A small part of the cooled carriers may recombine with the ground state through a fast channel with lifetime of $\sim 354 \text{ ps}$ while the rest of carriers undergo the intersystem crossing from singlet to triplet states ($\sim 2.8 \text{ ns}$). Carriers occupying the triplet states recombine with the ground state slowly due to the spin forbidden effect ($\sim 9.2 \mu\text{s}$).

Figure R37 (a) Differential transmission spectra ($\Delta T/T$) at different time delays. (b) The curves probed at 650 nm, and the inset is the enlarged onset curve. (c) Nanosecond-resolved TA spectra in the decay range up to 100 μ s. (d) Schematic model of triplet dynamics.

The discussion about transient absorption (TA) spectra was inserted into page S39-S41, and the Figure R37 is renamed as Figure S25.

Reviewers' comments:

Reviewer #1 (Remarks to the Author):

The authors have made a number of revisions to the supporting information document that accompanies their manuscript in addition to several revisions to the main text. Some of these revisions partially address my prior misgivings about what I see as the inadequately supported conclusions presented in their prior submission. Overall, however, I do not find the authors' arguments sufficiently compelling to validate the following two critical points: 1) the purported half-metallic character of their material and 2) the role of triplet states in determining the reactivity of their material toward hydrogen evolution. I have outlined my specific concerns about these two topical areas below. Based on these concerns, I cannot at this time recommend publication of the manuscript in its current form by Nature Communications.

1) The authors have collected what they describe as "spin resolved photoemission" data. As plotted by the authors, these data appear to show subtle differences between the traces labeled "minority" and "majority" in Figure S13. However, whether the differences in the data presentation necessarily confirm that the material the authors have studied truly demonstrates "half-metallic" character is another matter. I find two main questions that are left unaddressed by the authors in this regard: A) Couldn't these photoemission data be described equally well as the result of a distribution of trap states or defect states that extend into the gap of a semiconducting material rather than "metallic" states for the "majority" spin? B) Are the secondary electron cut-off energies for both spin-resolved PES spectra the same or do they shift slightly?

2) I find unconvincing the authors' discussion and analysis concerning their time-resolved spectroscopy measurements with regard to their conclusion that they observe triplet-mediated photocatalytic activity. I have outlined below several of my concerns about this data and what I believe to be a number of flaws in the authors' analysis.

A) Regarding the authors' time-resolved differential transmission data: As the authors have defined it, $dT/T > 0$ signals correspond to loss of ground state absorption (bleaching behavior) and $dT/T < 0$ corresponds to absorptive behavior. While the authors assign the positive feature that appears in their $t = 0.2$ ps dT/T spectrum from 500 nm to roughly 750 nm to bleaching of the ground state absorption, their photoluminescence trace exhibits a high energy edge of at 500 nm and a peak near 550 nm, indicating that one should expect the bleach of the ground state absorption signal to be at wavelengths shorter than 500 nm and not at wavelengths longer than 500 nm. Given these results I find the authors' assignment of this feature as the bleach of the ground state absorption to be unsupported by the data. From where else could this feature originate?

B) The authors observe that their differential transmission spectra evolve to an induced absorption signal in the 550 – 800 nm range that decays on the nanosecond timescale. How do these results compare with recent transient absorption studies over similar timescales and spectral windows from, for example, Godin et al. [J. Am. Chem. Soc., 2017, 139 (14), pp 5216–5224] and Corp et al. [J. Am. Chem. Soc., 2017, 139 (23), pp 7904–7912], which appear to observe charge carrier formation on these timescales? At later times the authors report a positive feature that they again assign (see question in Part A above) as a ground state bleach signal associated with triplets. My understanding is that the authors use the long decay time of their photoluminescence signal as the basis for their argument that singlet-triplet interconversion is important in this material. Considering the aforementioned transient absorption examples in the literature, might it not be equally likely that some form of long-lived or trapped charge carrier is involved? What additional evidence is there to support the role of the triplet excited state in this system? Do the authors see quenching of the delayed fluorescence in the presence of oxygen or some other triplet acceptor? Is it not possible that

this long-lived positive feature is associated with trapped charge carriers rather than bleaching of the ground state related to triplet excitons? The authors have also apparently plotted the units incorrectly on what is ostensibly the time-evolution of the spectral intensity at 650 nm in Figure R37b. Assumedly, the units here should be "ns" but they are plotted as "nm."

C) The conclusion that the authors draw from their time-resolved photoluminescence data that there is a singlet-triplet splitting energy of 70 meV in this material seems questionable. This is an extremely small singlet-triplet splitting, even for a charge transfer excitation. What evidence is there that the red-shift in the PL signal with time is not simply due to energy migration (either exciton diffusion or charge trapping/detrapping) within a disordered density of states that ultimately localizes on the lowest energy polymer chain over time? Is this longtime component quenched in the presence of molecular oxygen? Detecting a half-field light-induced EPR signal here for the $S = 1$ state would be convincing evidence that there is a significant triplet population.

Additionally, in my view, the authors present an inconsistent discussion of whether the purported half-metallic character of their material leads to new catalytically active sites or not. In key passages of the manuscript (e.g. abstract) the authors make the claim that "The introduced half-metallic features not only effectively facilitate carrier transfer but also provide more active sites for hydrogen evolution reaction." However, at other points in the manuscript the authors seem to suggest that their material acts to more efficiently funnel charge to the co-catalyst rather facilitating a higher active site density. Which is it?

Reviewer #3 (Remarks to the Author):

The authors basically have addressed all the concerns by the reviewers. The current version meets my expectation and now I recommend it publishing in Nature Communications.

**"Half-Metallic Carbon Nitride Nanosheets with Micro Grid
Mode Resonance Structure for Highly Efficient Photocatalytic
Hydrogen Evolution"**

ID: NCOMMS-17-30419B

Response to the report of the Reviewer 1

The authors have made a number of revisions to the supporting information document that accompanies their manuscript in addition to several revisions to the main text. Some of these revisions partially address my prior misgivings about what I see as the inadequately supported conclusions presented in their prior submission. Overall, however, I do not find the authors arguments sufficiently compelling to validate the following two critical points: 1) the purported half-metallic character of their material and 2) the role of triplet states in determining the reactivity of their material toward hydrogen evolution. I have outlined my specific concerns about these two topical areas below. Based on these concerns, I cannot at this time recommend publication of the manuscript in its current form by Nature Communications.

Comment 1. The authors have collected what they describe as “spin resolved photoemission” data. As plotted by the authors, these data appear to show subtle differences between the traces labeled “minority” and “majority” in Figure S13. However, whether the differences in the data presentation necessarily confirm that the material the authors have studied truly demonstrates “half-metallic” character is another matter. I find two main questions that are left unaddressed by the authors in

this regard:

A) Couldn't these photoemission data be described equally well as the result of a distribution of trap states or defect states that extend into the gap of a semiconducting material rather than "metallic" states for the "majority" spin?

Answer: Thanks for the reviewer's comments and suggestions. In order to clarify the physical origin of spin-resolved photoemission spectra, some additional experiments and theoretical calculations were carried out. In previous reports [Du, A. J et al, Phys. Rev. Lett. **108**, 197207 (2012); Lee, E. C. et al, Chem.- Eur. J. **16** 12141 (2010)], the half-metallic character is proved to be strongly related with introduction of radical C1 sites (C1 site, Figure 6a), which can inject a hole into original graphitic carbon nitride structure in one primitive cell, leading to spin polarization happened at neighboring N atoms.

To distinguish whether the magnetic contribution originates from defects or from radical C1 sites in pristine hm-C(CN)₃, theoretical predications are provided firstly. We investigate the relative stability of the defects by calculating the formation energy and hydrogenated energy using the following expression:

$$E_f = E_{\text{tot}}(\text{def}) + E_{\text{def}} - E_{\text{tot}}(\text{pristine}),$$

where $E_{\text{tot}}(\text{def})$ and $E_{\text{tot}}(\text{pristine})$ are the total energy of the (3×3) hm-C(CN)₃ supercells with and without defects, and E_{def} is the energy of pristine C, hydrogen or nitrogen, respectively. As a general feature, the calculated results in Table R1 disclose that all defects with larger positive formation energy cannot be unintentionally introduced, and those defects cannot be easily deactivated by hydrogenation treatment

either. This is because that the chemical bonds at defected region will be restructured to form new spin polarization, as shown in Figure R1, and those formed covalent bonds are too strong to unfold to react with hydrogen. On the contrary, reaction barrier calculated by climbing nudged elastic band (NEB) method [Henkelman, G. et al, J. Chem. Phys, **113**, 9978 (2000)] reveals that the radical C1 sites of pristine hm-C(CN)₃ can be easily deactivated by hydrogen after overcoming a 0.26 eV potential barrier, finally to achieve nonmagnetic structures with negative hydrogenation energy, as shown in Figure R1 (a) and (e). This table (Table R1) has been added in page 8 lines 164-165, page 8 lines 168-170, and in the supporting information (renamed as Table S1, page S57-58).

Table R1. The formation energy, hydrogenated energy and reaction potential barrier for hm-C(CN)₃ monolayer with different defects.

Types	Pristine	V1	V2	V3
Formation Energy	0.00 eV	4.77 eV	5.81 eV	1.75 eV
Hydrogenated Energy	-2.27 eV	1.23 eV	1.34 eV	0.24 eV
Reaction Barrier	0.26 eV	0.78 eV	0.82 eV	0.45 eV

Figure R1. Calculated spin densities of hm-C(CN)₃ monolayer with different defects before (a-d) and after (e-h) hydrogenation treatment. Gray, yellow and pink balls stand for C, N and H atoms, respectively.

Subsequently, we discuss the influence of hydrogenation on magnetic behavior in detail. The spin charge density between spin-up and spin-down ($\rho = \rho_{\downarrow} - \rho_{\uparrow}$) is symmetrically distributed in neighboring N atoms (yellow atom), as shown in Figure R1a. This is because the existing radical C sites (targeted by pink H atoms) can inject holes into this structure, and superfluous charges are redistributed in nearby N atoms.

The distribution of spin polarization can be affected by different defects, as shown in Figure R1b-1d, leading to decrease in magnetic moments (different from reviewer's speculation). After hydrogenation into those structures, the radical C sites are saturated by H atoms (pink atoms) to eliminate superfluous charges, which makes spin polarization and magnetic moments disappear simultaneously (Figure R1e). However, spin polarizations are completely unaffected at defect vicinity (marked by circles) (Figure R1f-1h). These results demonstrate that the spin polarizations induced by radical C sites are strongly associated with hydrogenation. This part (Figure R1) has been added in page 8 lines 164-165, page 8 lines 168-170, and in the supporting information (renamed as Figure S13, page S23-S24).

To more intuitively display contribution of hydrogenation, the magnetic moments of hm-C(CN)₃ monolayer with different number of V3 defects as functions of hydrogenation concentration are calculated and compared in Figure R2. We found that the magnetic moments of hm-C(CN)₃ monolayer without defects drop linearly with increasing hydrogenation concentration, and finally disappear completely, which is quite in agreement with calculated results in Figure R1a and R1e. However, when some V3 defects with the lowest formation energy are introduced intentionally, the original magnetic moment values are depressed, which is not benefit for half-metallic observation in spin resolved photoemission spectra. More importantly, the hydrogenation cannot make magnetism originated from defects be eliminated completely. Note that this behavior becomes more remarkable especially for higher defect concentration. This evidences that the magnetic origin can be identified via

hydrogenation treatment. This part (Figure R2) has been added in page 8 lines 164-165, page 8 lines 168-170, and in the supporting information (renamed as Figure S14, page S25).

Figure R2. Calculated magnetic moment as a function of hydrogenation concentration in $\text{hm-C}(\text{CN})_3$ monolayer with different V3 defect concentrations.

Next, we experimentally distinguish the physical mechanism of spin resolved photoemission spectra via hydrogenation treatment. The $\text{hm-C}(\text{CN})_3$ nanosheets were placed on an alumina boat and inserted into a quartz tube center of a horizontal pipe furnace. Before heating, the system was purged with 517 standard cubic centimeter per minute (sccm) high-purity argon (Ar 99.999%) for 30 min. Then, the quartz tube was evacuated to a pressure of 1×10^{-5} mbar by a mechanical pump for the duration of the reaction. After that, the furnace was heated at a heating rate of $10 \text{ }^\circ\text{C min}^{-1}$ to

specific temperature of 300°C. It was kept at this temperature for 2.5 hour with a mixture of Ar (98%) and H₂ (2%) flow of 150 sccm. After the system cooled down to room temperature, a hydrogen-passivated product was obtained on the alumina boat. To confirm the hydrogenation process, the ¹³C NMR spectra of hm-C(CN)₃ sample before and after hydrogenation were measured and shown in Figure R3. It is important to note that only three typical peaks are observed in consistence with structural symmetry, which also indicates that no obvious defected structures exist in our samples. This is because the hm-C(CN)₃ structures will be destructed, and some new C sites will be visible in NMR spectra, if some defects are introduced unintentionally. In addition, the peak 3 originated from radical C sites slightly shifts after hydrogenation treatment, further confirming that hydrogen deactivation occurs at radical C sites and no additional defect structures are introduced. This part (Figure R3) has been added in page 8 lines 170-174, and in the supporting information (renamed as Figure S15, page S26 and page S28).

Figure R3. ¹³C NMR spectra of R@hm-C(CN)₃ nanosheets before and after

hydrogenation.

The spin resolved photoemission spectra (PES) of hm-C(CN)₃ sample after hydrogenation treatment was recorded and compared to that of before hydrogenation treatment, as shown in Figure R4. The spectral difference between majority and minority state disappears, in good agreement with theoretical prediction. This further confirms that the half-metallic feature does not originate from defected structures but from radical C sites, because DFT calculations show that the spin polarization induced by some defects cannot be completely eliminated in hm-C(CN)₃ sample, and that the difference in spin resolved photoemission spectra (PES) should remain unquenched after hydrogenation treatment. This part (Figure R4) has been added in page 8 lines 174-176, and in the supporting information (renamed as Figure S16, page S28-S29).

Figure R4. The spin resolved photoemission spectra of hm-C(CN)₃ nanosheets before and after hydrogenation.

To further confirm our assertions, the spin-resolved DOS for hm-C(CN)₃ monolayer with different defects were calculated and shown in Figure R5. The results indicate that the spin splitting of DOS can be affected by defects, resulting in changes in spin DOS feature [DOS difference between majority-spin (down) and minority-spin (up)]. The notable difference between experimental PES spectra and calculated spin-resolved DOS in hm-C(CN)₃ monolayer with different defects further evidences that the difference in PES spectra does not result from defects. It is important to note that calculated DOS for pristine hm-C(CN)₃ is much similar to experimental PES spectra, as shown in Figure R5a, although the spectral difference between majority and minority spins is not so large as the spin resolved DOS by DFT. This part (Figure R5) has been added in page 8 lines 174-176 and in the supporting information (renamed as Figure S17, page S30).

Figure R5. Calculated spin-resolved DOSs and the difference for hm-C(CN)₃ monolayer with and without defects.

To find the fundamental causes that underlie the spectral differences between PES spectra and spin-resolved DOSs, the polymerization process and XRD patterns of our samples should be discussed in detail. As we all known, experimental samples cannot be as perfect as theoretical simulation, and lattice distortion and deformation cannot be avoided in sample preparation, especially for two dimensional materials [such MoS₂ or Graphene, see Nano. Lett. **10** 4074 (2010); Nat. Mater. **17** 129 (2018); Nat. Nanotech. **13** 152 (2018)]. In addition, the broad XRD peak implies that deformation induced by residual strain also takes place in our samples. Considering this physical truth, spin-resolved DOS of hm-C(CN)₃ monolayer with different deformations were calculated and shown in Figure R6. The calculated results show that the spin splitting in DOS decreases with increasing structural deformation and finally tends to measured PES spectra. This is because spin polarization can be regulated by structural deformation [see our previous report, Appl. Phys. Lett. 106, 132406 (2015)]. In addition, the stimulated XRD patterns in Figure R7 indicate that the broad XRD peak can be assigned to a combination of contributions from hm-C(CN)₃ nanosheets with and without deformation. In this context, the structural deformation induced by residual strain can modify the PES feature to a certain extent. Taking this factor into consideration, the differences between experimental PES spectra and DFT predicated DOS can be satisfactorily explained. This two parts (Figure R6 and R7) have been added in page 8 lines 176-179, and in the supporting information (renamed as Figure

S18 and Figure S19, pages S31-S33).

Figure R6. Calculated spin-resolved DOS for hm-C(CN)₃ monolayer with different deformation.

Figure R7. The experimental and the theoretical XRD patterns with different deformations.

Based on the physical truth that the differences between PES spectra and calculated spin-resolved DOS are related with structural deformation induced by residual strain but not defects, the calculated DOSs acquired on combined contributions from hm-C(CN)₃ with and without deformation are in good agreement with measured PES spectra. These results also indicate that the half-metallic feature

only originates from pristine hm-C(CN)₃. This part is inserted into page S31 and S32.

B) Are the secondary electron cut-off energies for both spin-resolved PES spectra the same or do they shift slightly?

Answer: The secondary electron cut-off energies for both spin-up and spin-down electrons are the same. In the measurement, the energy conservation is used as

$$E_k = \hbar\nu - E_\phi - E_b,$$

where $\hbar\nu$, E_ϕ , E_b , E_k is the exciting light energy, work function, electron binding energy and photoelectron kinetic energy, respectively. We use hemisphere analyzer to extract the kinetic energy of the photoelectrons, while the $\hbar\nu$ and E_ϕ are constants for one measurement and the same for both spin-up and spin-down electrons. And then the binding energy of the electrons can be calculated. Here the spins do not affect the binding energy.

After the hemisphere analyzer, the photoelectrons are accelerated at 30 keV to bombard on an Au target. Because of the spin-orbit coupling of Au, the spin-up and spin-down electrons will be scattered to different directions, and then detected by two electron detectors respectively. This is how the spin-resolved PES spectra are obtained. The schematic sketch of spin-resolved PES measurement is shown in Figure R8.

Figure R8. The schematic sketch of spin-resolved PES measurement.

2) I find unconvincing the authors' discussion and analysis concerning their time-resolved spectroscopy measurements with regard to their conclusion that they observe triplet-mediated photocatalytic activity. I have outlined below several of my concerns about this data and what I believe to be a number of flaws in the authors' analysis.

A) Regarding the authors' time-resolved differential transmission data: As the authors have defined it, $dT/T > 0$ signals correspond to loss of ground state absorption (bleaching behavior) and $dT/T < 0$ corresponds to absorptive behavior. While the authors assign the positive feature that appears in their $t = 0.2$ ps dT/T spectrum from 500 nm to roughly 750 nm to bleaching of the ground state absorption, their photoluminescence trace exhibits a high energy edge of at 500 nm and a peak near 550 nm, indicating that one should expect the bleach of the ground state absorption

signal to be at wavelengths shorter than 500 nm and not at wavelengths longer than 500 nm. Given these results I find the authors' assignment of this feature as the bleach of the ground state absorption to be unsupported by the data. From where else could this feature originate?

Answer: Thanks for the reviewer's comments and suggestions. We completely agree with reviewer's comment that this broad absorption peak from 550 to 800 nm cannot be simply attributed to ground-state bleaching signal. Similar positive features have been observed in previous studies and are assigned to photogenerated holes or electrons, or electron-hole pairs [J. Am. Chem. Soc. 139, 7904 (2017)]. To investigate its origin, the pump power-dependence of this induced absorption is shown in Figure R9a. In a considerable range of laser power, we find that the decay dynamics of the induced TA signal remains unchanged, which is obviously different from the previous reported no-geminate charge recombination [J. Am. Chem. Soc. 139, 7904 (2017); J. Am. Chem. Soc. 139, 5216 (2017)]. On the contrary, this significant pump power-independent TA decay behavior is often regarded as quasi-monomolecular decay dynamics associated with neutral electron-hole pair. To further validate absorption assignment of excited-state electron-hole, we utilize electron-phonon interaction to eliminate the contribution from electron-hole pairs to broad absorption peaks at 550-800 nm. Increasing measurement temperature is a powerful method to identify the absorption spectra associated with the generation and disappearance of electron-hole pairs. Figure R9b shows the changes in TA signal at 0.2 ps at different temperatures, which indicates that the broad absorption is quenched at higher

temperature while the photoinduced bleaching signal keeps unperturbed. This temperature-dependent absorption suggests that a portion of the positive feature in the TA starting from ca. 550 to 800 nm corresponds to photogenerated electron-holes pairs. Then, we increase the temperature to 318 K to exclude influence of electron-hole pairs on our assignment of long-lifetime TA components, and the induced absorption at different time delays were recorded again and shown in Figure R9c. Interestingly, the late-stage TA signals from 20 ps to 3.9 ns is nearly unperturbed, which is quite in agreement with results observed at 298 K (Figure S31a). This insensitivity to the temperature is an evidence of singlet-triplet conversion occurring in our materials, which is consisted with analysis about transient fluorescence (Figure S30) and similar to those recently described by Chen et al [J. Phys. Chem. C 121 12972 (2017)]. In conclusion, the broad TA signals (550-700 nm) emerged at the time decays < 20 ps can be attributed to photogenerated electron-hole pairs and can be eliminated by electron-phonon interaction, but this early stage signals cannot affect our assignment about singlet-triplet conversion at late-stage signals. This part (Figure R9) has been added in page 16 lines 342-343, and in the supporting information (renamed as Figure S32, page S51-S53).

Figure R9. Ultrafast carrier dynamics. (a) The normalized curves probed at 650 nm with different pump powers. (b) The transmission spectra acquired at 0.2 ps for different measuring temperatures. (c) The differential transmission spectra at different time delays acquired at 318 K. (d) The kinetic curves probed at 650 nm recorded from hm-C(CN)₃ nanosheets before and after hydrogenation and dispersed in ethanol with dissolved oxygen.

B) The authors observe that their differential transmission spectra evolve to an induced absorption signal in the 550 – 800 nm range that decays on the nanosecond timescale. How do these results compare with recent transient absorption studies over similar timescales and spectral windows from, for example, Godin et al. [J. Am. Chem.

Soc., 2017, 139 (14), pp 5216–5224] and Corp et al. [J. Am. Chem. Soc., 2017, 139 (23), pp 7904–7912], which appear to observe charge carrier formation on these timescales? At later times the authors report a positive feature that they again assign (see question in Part A above) as a ground state bleach signal associated with triplets. My understanding is that the authors use the long decay time of their photoluminescence signal as the basis for their argument that singlet-triplet interconversion is important in this material. Considering the aforementioned transient absorption examples in the literature, might it not be equally likely that some form of long-lived or trapped charge carrier is involved? What additional evidence is there to support the role of the triplet excited state in this system? Do the authors see quenching of the delayed fluorescence in the presence of oxygen or some other triplet acceptor? Is it not possible that this long-lived positive feature is associated with trapped charge carriers rather than bleaching of the ground state related to triplet excitons? The authors have also apparently plotted the units incorrectly on what is ostensibly the time-evolution of the spectral intensity at 650 nm in Figure R37b. Assumedly, the units here should be “ns” but they are plotted as “nm.”

Answer: Thanks for the reviewer's comments and suggestions. We agree that the long-lived positive feature also can be induced by charge carrier formation or trapped charge carrier. The pump power-dependence of the induced absorption is a powerful tool to identify charge carrier or singlet-triplet conversion. As shown in Figure R9a, the excited-state absorption decay rates in our material exhibit marked power independence, contrary to those recently charge carrier behavior described by Godin

et al. [J. Am. Chem. Soc., 2017, 139 (14), pp 5216–5224] and Corp et al. [J. Am. Chem. Soc., 2017, 139 (23), pp 7904–7912]. This conclusion indicates that TA spectra in our reported system have totally different physical mechanism. The assignment of the long-lived component to triplet generation is further confirmed by an oxygen quenching measurement, i.e., by introducing oxygen into the dispersion of hm-C(CN)₃ nanosheets. The lifetime of the long-lived component is significantly shortened (Figure R9d), indicating a strong interaction between oxygen and the excited triplet states. These results strongly support the formation of triplet state [J. Am. Chem. Soc. 107, 6726 (1985)]. The average lifetime of the long-lived component is shortened to 1.1 μs in the presence of oxygen, suggesting that ~75% of excited triplet states may be involved in the generation of singlet oxygen. In addition, an obvious quenching behavior was also observed in hm-C(CN)₃ with hydrogenation treatment via which the half-metallic feature disappears (Figure R4), similar to oxygen quenching curve. This confirms that the intrinsic half-metallicity plays a critical role in generation of excited triplet states. Based on the above results, we can safely describe the dynamics of photoexcited carriers in hm-C(CN)₃ nanosheets. Following the subpicosecond thermalization process to eliminate electron-hole pairs, a large portion of carriers undergo the intersystem crossing to form the triplet carriers. The long-lived triplets are affected by hydrogenation, which is critical for the hydrogen evolution reaction. In addition, the erroneous units have been corrected as shown in Figure S31. This part has been added in page 16 lines 343-346, line 349, and in the supporting information (page S53).

C) The conclusion that the authors draw from their time-resolved photoluminescence data that there is a singlet-triplet splitting energy of 70 meV in this material seems questionable. This is an extremely small singlet-triplet splitting, even for a charge transfer excitation. What evidence is there that the red-shift in the PL signal with time is not simply due to energy migration (either exciton diffusion or charge trapping/detrapping) within a disordered density of states that ultimately localizes on the lowest energy polymer chain over time? Is this longtime component quenched in the presence of molecular oxygen? Detecting a half-field light-induced EPR signal here for the $S = 1$ state would be convincing evidence that there is a significant triplet population.

Answer: Thanks for the reviewer's comments. The singlet-triplet splitting energy of 70 meV is acquired from time-resolved photoluminescence (PL) spectra, which is close to previous reports, such as 83 meV in carbazolyl dicyanobenzene system [Nature, 492, 235 (2012)]; 30-80 meV in polymer structure [Adv. Mater. 29, 1604223 (2017)]. Considering the aforementioned transient absorption of hm-C(CN)₃ nanosheets at different pump powers (Figure R9a), in oxygen solution and after hydrogenation (Figure R9d), we conclude that the shift in the PL signal cannot originate from energy migration (either exciton diffusion or charge trapping/detrapping). To further confirm our assertion, the time-resolved PL spectra of hm-C(CN)₃ nanosheets in oxygen solution were measured again and shown in Figure R10a. We can see that the shifted PL component is significantly reduced to 532 nm from 551 nm, because the excited triplet states are consumed for generation of singlet

oxygen. In addition, similar behavior is also observed in hydrogenated samples (Figure R10b), which indicates that singlet-triplet conversion is depressed due to disappearance of half-metallic feature. In addition, the lifetimes of the PL decay in oxygen solution and after hydrogenation are shortened from 7.9 μs to 31 ns or 45 ns, respectively. This conclusion is in accordance with that of TA spectra. This part (Figure R10 has been added in page 16 lines 343-346, line 349, and in the supporting information (renamed as Figure S33, page S54-S56).

Figure R10. Time-resolved photoluminescence spectra of hm-C(CN)₃ nanosheets in the presence of concentrated dissolved oxygen (a) and after hydrogenation (b), respectively. The photoluminescence decay curves (c) and the half-field light-induced

electron spin resonance (ESR) spectra (d) of hm-C(CN)₃ nanosheets in the presence of concentrated dissolved oxygen and after hydrogenation.

To further confirm the triplet generation, the half-field light-induced electron spin resonance (ESR) spectra, a convincing evidence to identify the species of generated spin triplet state, were recorded and shown in Figure R10d. The ESR spectra under light irradiation ($\lambda > 600$ nm) show very similar signals for pristine hm-C(CN)₃ sample, with a Lande factor $g=2.11$. Besides this large signal which corresponds to the $\Delta M_s = \pm 1$ allowed transition, the pristine sample exhibits a weak signals (enlarged by 600 times) located at lower field values (corresponding to the $\Delta M_s = \pm 2$ forbidden transition). These can be assigned to forbidden transitions between the singlet electronic state of an exchange-coupled neighboring radical C sites and the $M_s = \pm 1$ components of the triplet state. When it is taken into account that each of these transitions is located at $2|J|/g\beta$ to lower field of the allowed transition, exchange parameters $|J|$ can be calculated as 0.11 cm^{-1} . When the oxygen is introduced into the solution of hm-C(CN)₃ nanosheets, the signals associated with the triplet $\Delta M_s = \pm 1$ and $\Delta M_s = \pm 2$ transitions are quenched because excited triplet states are used to produce singlet oxygen. In addition, when the lone pair of electrons in hm-C(CN)₃ nanosheet are saturated by hydrogenation treatment, the ESR signals are weakened and singlet-triplet conversion also disappears. These results are well in consistence with the TA and the time-resolved PL spectra. This part (Figure R10) has been added in page 16 lines 350-352, and in the supporting information (renamed as Figure S33,

page S54-S56).

Additionally, in my view, the authors present an inconsistent discussion of whether the purported half-metallic character of their material leads to new catalytically active sites or not. In key passages of the manuscript (e.g. abstract) the authors make the claim that “The introduced half-metallic features not only effectively facilitate carrier transfer but also provide more active sites for hydrogen evolution reaction.” However, at other points in the manuscript the authors seem to suggest that their material acts to more efficiently funnel charge to the co-catalyst rather facilitating a higher active site density. Which is it?

Answer: Thanks for the reviewer's comments and suggestions. In the abstract and the introduction section of our manuscript, we state that the introduced half-metallic features not only effectively facilitate carrier transfer but also provide more active sites for hydrogen evolution reaction. In the following section, we present a consistent discussion to support our statement. When the spin resonance spectra of hm-C(CN)₃ sample are talked about in page 7 (Figure 1f and XPS spectra in Figure S7), we claim that the superfluous C atoms (the origin of half-metallicity) can be used as additional active sites. In the discussion about the spin-resolved total density of state (TDOS) in Figure 2d in page 8, it is again stressed that the hm-C(CN)₃ surface is fully filled with metallic spin-down states, which is more advantageous to electron donating of a photocatalyst. Here, we mean that half-metallicity can facilitate carrier transfer, which

is also confirmed by the electrical conductivity in Figure 5a (page 11). In addition, the contribution of half-metallic feature induced by those radical C atoms to photocatalytic hydrogen evolution by singlet-triplet conversion is further discussed in page 16. More importantly, those radical C atoms (such as C1 site in Figure 6a) acted as active sites also take part into the hydrogen evolution reaction, as discussion about Figure 6 in page 14-15. Based on above analyses, we get the conclusion that the introduced half-metallic features not only effectively facilitate carrier transfer but also provide more active sites for hydrogen evolution reaction. If the statement is understood by the reviewer as *'their material acts to more efficiently funnel charge to the co-catalyst rather facilitating a higher active site density'*, we are very sorry to make such a confusion. This sentence has been rewritten as "The hm-C(CN)₃ surface is fully filled with metallic spin-down states (see inset), which is more advantageous to carrier transfer " to make it understood properly, please see page 9 lines 183-185.

REVIEWERS' COMMENTS:

Reviewer #1 (Remarks to the Author):

The authors have addressed several of my primary misgivings about unsupported conclusions in their manuscript by, for example, providing additional spin resolved photoemission measurements and light-induced half field EPR data. While I still have severe reservations about what I consider to be an over-interpretation of the author's observations, given the structural ambiguity that is inherent to this material, I am reluctantly prepared to let the scientific community judge for themselves. Overall, the authors present an interesting perspective on the behavior of a material that is of broad interest to chemists, physicists, chemical engineers, and materials scientists. As such, the topical area and the catalogue of measured data contained within the manuscript seem appropriate for the audience of Nature Communications. If there is sufficient enthusiasm from the remaining reviewers of this work for publishing in Nature Communications, I will defer to their expert opinions.

**"Half-Metallic Carbon Nitride Nanosheets with Micro Grid
Mode Resonance Structure for Highly Efficient Photocatalytic
Hydrogen Evolution"**

ID: NCOMMS-17-30419C-Z

Response to the report of the Reviewer 1

The authors have addressed several of my primary misgivings about unsupported conclusions in their manuscript by, for example, providing additional spin resolved photoemission measurements and light-induced half field EPR data. While I still have severe reservations about what I consider to be an over-interpretation of the author's observations, given the structural ambiguity that is inherent to this material, I am reluctantly prepared to let the scientific community judge for themselves. Overall, the authors present an interesting perspective on the behavior of a material that is of broad interest to chemists, physicists, chemical engineers, and materials scientists. As such, the topical area and the catalogue of measured data contained within the manuscript seem appropriate for the audience of Nature Communications. If there is sufficient enthusiasm from the remaining reviewers of this work for publishing in Nature Communications, I will defer to their expert opinions.

Answer: Thanks for the reviewer's kind comments and we highly appreciate the reviewer's rigorous attitude to our work. With help of the reviewer, our work has been greatly improved to meet the highly scientific requirements of Nature Communications.

Although the structural ambiguity is inherent to carbon nitride material, here, the prepared half-metallic carbon nitride material in our work is proved to be of unambiguous structure and has high photocatalytic efficiency for hydrogen evolution, based on theoretical calculations and solid experimental evidences such as spin resolved photoemission, transient absorption and transient fluorescence. And, the hydrogen evolution reaction has been systematically confirmed by theoretical simulation (Fig. 6) and spectral evidences (supplementary Figure 29-33). Thus, all the scientific conclusions in our work, e.g., half-metallicity and photocatalytic activity, are drawn on solid evidences.

Since the reviewer agrees with the remaining reviewers who have recommended publishing our work at present form, we believe that the present version of our manuscript is suitable for publishing in Nature Communications.

Thanks again for the reviewer's help!